# Dynamical Modeling of Behaviorally Relevant Spatiotemporal Patterns in Neural Imaging Data

Mohammad Hosseini [1]   Maryam M. Shanechi [1 2 3]

## Abstract

High-dimensional imaging of neural activity, such as widefield calcium and functional ultrasound imaging, provide a rich source of information for understanding the relationship between brain activity and behavior. Accurately modeling neural dynamics in these modalities is crucial for understanding this relationship but is hindered by the high-dimensionality, complex spatiotemporal dependencies, and prevalent behaviorally irrelevant dynamics in these modalities. Existing dynamical models often employ preprocessing steps to obtain low-dimensional representations from neural image modalities. However, this process can discard behaviorally relevant information and miss spatiotemporal structure. We propose SBIND, a novel data-driven deep learning framework to model spatiotemporal dependencies in  neural images and disentangle their behaviorally relevant dynamics from other neural dynamics. We validate SBIND on widefield imaging datasets, and show its extension to functional ultrasound imaging, a recent modality whose dynamical modeling has largely remained unexplored. We find that our model effectively identifies both local and long-range spatial dependencies across the brain while also dissociating behaviorally relevant neural dynamics. Doing so, SBIND outperforms existing models in neural-behavioral prediction. Overall, SBIND provides a versatile tool for investigating the neural mechanisms underlying behavior using imaging modalities.

[1]Electrical and Computer Engineering, Viterbi School of Engineering, University of Southern California (USC), Los Angeles, CA, USA [2]Computer Science, Viterbi School of Engineering, USC, Los Angeles, CA, USA [3]Biomedical Engineering, Viterbi School of Engineering, USC, Los Angeles, CA, USA. Correspondence to: Maryam M. Shanechi <shanechi@usc.edu>.

*Proceedings of the $42^{nd}$ International Conference on Machine Learning*, Vancouver, Canada. PMLR 267, 2025. Copyright 2025 by the author(s).

## 1. Introduction

Recent advances in neuroimaging techniques, such as widefield calcium and functional ultrasound imaging, have provided unprecedented access to high-dimensional neural data that can measure the brain at larger spatial scales than traditional electrophysiological modalities (Macé et al., 2011; Musall et al., 2019; Benisty et al., 2022). These imaging techniques are increasingly central to modern neuroscience, offering complementary insights to electrophysiological recordings by enabling the study of large-scale network dynamics, distributed neural representations, and functional connectivity, which is critical for understanding cognition and complex behavior (Cardin et al., 2020; Mace et al., 2018). Widefield calcium imaging utilizes fluorescent indicators to monitor calcium influx in neurons, capturing mesoscale neural activity across a large expanse of the cortical surface with relatively high temporal resolution compared to other neural imaging modalities (Musall et al., 2019; Ren & Komiyama, 2021; Nietz et al., 2023). Functional ultrasound, on the other hand, detects changes in cerebral blood volume, which are correlated with both single-neuron activity and local field potentials, offering wider spatial coverage and a less invasive approach compared to electrophysiological implants, with great potential for brain-computer interfaces (BCIs) (Nunez-Elizalde et al., 2022; Griggs et al., 2024; Rabut et al., 2024).

Despite the rich spatial and temporal information that these imaging modalities can provide as well as their potential application in BCIs, fully capturing their complex spatiotemporal dynamics and their link to observed behavior has remained elusive due to several challenges (Dinsdale et al., 2022). Specifically, these modalities are high-dimensional and have spatiotemporal patterns that are complex and include both local and global dependencies. Furthermore, these patterns also contain prevalent behavior-irrelevant components. Hence, developing new methods that address the distinct challenges of neural imaging data to accurately model such datasets is crucial to both investigate brain-behavior links and enable their use as new modalities in BCIs (Norman et al., 2021; Griggs et al., 2024).

To analyze these high-dimensional neural imaging datasets, existing work often employ an initial dimensionality re-

duction step in the form of preprocessing before behavior decoding or latent state modeling. This step may utilize unsupervised methods such as principal component analysis (PCA) (Musall et al., 2019), or rely on pre-defined regions of interest (ROIs) in the brain (Saxena et al., 2020; Wang et al., 2020). These preprocessed low-dimensional features are then used in modeling, for example, to investigate neural-behavioral relationships in widefield calcium imaging data (Batty et al., 2019; Whiteway et al., 2021; Benisty et al., 2024; Wang et al., 2024) or to decode movement intentions from functional ultrasound data (Griggs et al., 2024). While computationally efficient, these preprocessing approaches may discard spatiotemporal information and inadvertently remove behaviorally relevant dynamics, limiting the ability to fully capture the complex spatiotemporal patterns that link neural activity to behavior.

Further complicating this problem, neural recordings often contain a vast amount of information that does not relate to a specific behavior of interest (Sani et al., 2021; Stringer et al., 2019; de Vries et al., 2020; Hasnain et al., 2023; Vahidi et al., 2024). This is especially the case for neural imaging modalities because they cover a large spatial scale in the brain, including many regions (Musall et al., 2019; Whiteway et al., 2021). Thus, a challenge in modeling neural activity is to disentangle behaviorally relevant neural dynamics from other ongoing processes in the brain (Sani et al., 2021). Indeed, traditional unsupervised learning methods for modeling neural activity may not optimally extract the behaviorally relevant neural components and may mix them with other neural components. To address this challenge, recent studies have jointly used neural and behavioral data during model training, leading to more accurate inference of behaviorally relevant neural dynamics (Sani et al., 2021; Hurwitz et al., 2021; Schneider et al., 2023; Gondur et al., 2024; Sani et al., 2024; Wang et al., 2024; Oganesian et al., 2024; Vahidi et al., 2025). Some of these works have employed dynamical models, which model the temporal evolution of time-series data in terms of a latent state. However, neural-behavioral models have either not focused on neural imaging modalities or used a preprocessing step as noted above.

**Contributions** To address the above limitations, we propose SBIND (Spatiotemporal modeling of Behavior in Imaging Neural Data), a novel deep learning framework for dynamical modeling of complex local and global spatiotemporal patterns in neural imaging data and disentangling their behaviorally relevant components. SBIND utilizes convolutional recurrent neural networks (ConvRNNs) to capture local short-range spatiotemporal dependencies and combines them with self-attention that is integrated into the dynamics to capture global spatiotemporal dependencies in the original image data. Moreover, to disentangle behaviorally relevant dynamics in these local and global patterns,

we devise a two-phase learning approach, where one ConvRNN first learns the behaviorally relevant dynamics, and then a subsequent ConvRNN captures other neural dynamics. To our knowledge, SBIND is the first neural-behavioral dynamical model to learn directly from raw widefield and functional ultrasound imaging data, without relying on preprocessing. Also, our work demonstrates the first dynamical latent modeling of functional ultrasound modality. We show that SBIND achieves superior performance in both behavior decoding and neural prediction for widefield imaging and functional ultrasound imaging data compared to other neural-behavioral models. Also, SBIND can learn neural dynamics in datasets with various behavior distributions, such as continuous, categorical, and intermittently recorded behavior.

## 2. Related Work

When working with widefield imaging data, preprocessing in the form of dimensionality reduction is almost always employed to obtain low-dimensional representations from raw widefield images. These methods can be broadly categorized into two families: (1) *Unsupervised dimensionality reduction*, where methods like PCA or Independent Component Analysis are used to reduce the data dimensionality before further neural modeling (Musall et al., 2019; Nietz et al., 2023; West et al., 2024; Scaglione et al., 2024). However, these unsupervised techniques can discard the spatial dependencies between brain regions inherent in widefield data. (2) *ROI-based methods*, which leverage brain atlases to incorporate spatial information during dimensionality reduction (Mishne et al., 2018; Liu et al., 2019; Wang et al., 2020). A popular example in the second family is LocaNMF (Saxena et al., 2020), which is frequently used in modeling widefield data (Batty et al., 2019; Wang et al., 2024). In functional ultrasound imaging (fUSI), while the literature is more limited, unsupervised techniques such as PCA have been used before decoding movement intentions from these neural images (Norman et al., 2021; Griggs et al., 2024).

These dimensionality reduction methods are often optimized to extract features that capture maximum variance in neural images, sometimes incorporating auxiliary losses to account for spatial information, such as in LocaNMF. Current methods then utilize these extracted features for dynamical modeling (Batty et al., 2019; Wang et al., 2024; Benisty et al., 2024; Karniol-Tambour et al., 2024) . However, a common drawback of this approach for dynamical modeling is that spatial information may be lost during feature extraction; furthermore, dimensionality reduction is performed without considering the behavior of interest, which can lead to missing crucial behavior-related information in neural imaging data. Our approach addresses these limitations by 1) directly modeling the raw neural image data, capturing both local

and global spatiotemporal dependencies through ConvRNN and self-attention mechanisms, and 2) learning the dynamical model jointly with neural images and behavioral data to disentangle behaviorally relevant image dynamics.

Outside the neural imaging domain and primarily for modeling electrophysiological neural modalities, various unsupervised learning methods have been developed to capture neural dynamics agnostic to behavior (Pandarinath et al., 2018; Le & Shlizerman, 2022; Wang et al., 2023; Li et al., 2024; Lu et al., 2025; Abbaspourazad et al., 2024). For example, STNDT (Le & Shlizerman, 2022) models spatiotemporal structure in Poisson-distributed spiking activity using a Transformer architecture. However, these unsupervised methods neither incorporate image priors nor aim to dissociate behaviorally relevant dynamics.

To learn behaviorally relevant neural dynamics, recent studies have developed neural-behavioral models that jointly consider neural activity and behavior during learning (Sani et al., 2021; Hurwitz et al., 2021; Schneider et al., 2023; Sani et al., 2024; Wang et al., 2024; Oganesian et al., 2024). A method termed PSID (Sani et al., 2021) dissociates behaviorally relevant neural dynamics within a linear dynamical system model. A recent method named CEBRA (Schneider et al., 2023) extracts latent embeddings using a learning framework that incorporates behavioral supervision through a contrastive loss. Another recent method termed DPAD (Sani et al., 2024) learns a dynamical model in the form of a two-section RNN that dissociates behaviorally relevant dynamics by incorporating an optimization stage focused solely on behavior prediction, while having subsequent stages learn other neural dynamics. Other methods such as BeNeDiff (Wang et al., 2024), TNDM (Hurwitz et al., 2021), and DFINE (Abbaspourazad et al., 2024) use a combined loss that incorporates both behavior and neural reconstruction to fit their dynamics. There are also multimodal methods such as mm-GP-VAE (Gondur et al., 2024) that fuse neural and behavioral data for behavior reconstruction, unlike the above neural-behavioral methods that only use the neural data for latent and behavior inference. Although effective, these methods either do not explicitly consider an image prior for the observed neural activity or do not directly address neural imaging data. Therefore, they may struggle in learning spatial information when raw image data is passed to the model.

Our method also jointly considers the neural-behavioral data during model training and disentangles behaviorally relevant neural dynamics. However, unlike the above neural-behavioral approaches that are not designed for raw neural images, our method integrates spatial priors directly into the model to capture both local and global information from raw neural images with image distributions. We show that this leads to more accurate neural-behavioral prediction for

neural imaging modalities.

## 3. Methods

### 3.1. SBIND Model

**Problem Formulation.** Figure 1 demonstrates the SBIND architecture. Neural activity ($\mathbf{Y}_k \in \mathbb{R}^{n_y \times H \times W}$) and simultaneously recorded behavior ($\mathbf{z}_k \in \mathbb{R}^{n_z}$) are modeled as observations generated by a dynamical system with latent states ($\mathbf{X}_k^g \in \mathbb{R}^{n_x \times H' \times W'}$). The generative model is defined as:

$$\begin{cases} \mathbf{X}_{k+1}^g &= \boldsymbol{f}_A^g(\mathbf{X}_k^g) + \mathbf{w}_k \\ \mathbf{Y}_k &= \boldsymbol{C}^g(\mathbf{X}_k^g) + \mathbf{v}_k \\ \mathbf{z}_k &= \boldsymbol{D}^g(\mathbf{X}_k^g) + \epsilon_k \end{cases} \quad (1)$$

where $\mathbf{X}_k^g$ represents the latent state at time $k$, modeled as a spatiotemporal representation in the form of a 3 dimensional (3D) latent volume characterized by its depth (number of channels, $n_x$), height ($H'$), and width ($W'$). $\mathbf{w}_k \in \mathbb{R}^{n_x \times H' \times W'}$ represents the noise affecting the latent state dynamics, and $\mathbf{v}_k \in \mathbb{R}^{n_y \times H \times W}$ and $\epsilon_k \in \mathbb{R}^{n_z}$ are the observation noises for neural images and behavior, respectively. Here, $H$ and $W$ are the height and width of the input neural images, $n_y$ is the number of input image channels ($n_y = 1$ in all the experiments, but here included for generality of the formulation), and $n_z$ is the dimensionality of the behavior vector.

Given this dynamical system, we can infer the latent state from neural observations $\mathbf{Y}_k$ using an RNN as follows:

$$\begin{cases} \mathbf{X}_{k+1} &= \boldsymbol{f}_A(\mathbf{X}_k) + \boldsymbol{K}(\mathbf{Y}_k) \\ \hat{\mathbf{Y}}_k &= \boldsymbol{C}(\mathbf{X}_k) \\ \hat{\mathbf{z}}_k &= \boldsymbol{D}(\mathbf{X}_k) \end{cases} \quad (2)$$

The RNN is parameterized by $\boldsymbol{f}_A$ (recurrence) and $\boldsymbol{K}$ (encoder); it estimates the latent state $\mathbf{X}_{k+1}$, given the past neural images $\{\mathbf{Y}_1, \ldots, \mathbf{Y}_k\}$. This latent state encapsulates all observed information up to time $k$. The predicted neural image at time $k$, $\hat{\mathbf{Y}}_k$, is derived from $\mathbf{X}_k$ via the neural decoder $\boldsymbol{C}(\cdot)$, while the decoded behavior $\hat{\mathbf{z}}_k$ is obtained using the behavior decoder $\boldsymbol{D}(\cdot)$.

**Model Parameterization and Design.** Our model is constructed using four key mappings that serve distinct roles in capturing the relationship between neural images and behavior: $\boldsymbol{f}_A(\cdot)$ describes the latent state recursion. the encoder, $\boldsymbol{K}(\cdot)$, maps the observed neural images into this latent representation. The decoders, $\boldsymbol{C}(\cdot)$ and $\boldsymbol{D}(\cdot)$, map the latent state to the corresponding neural and behavior observation spaces, respectively.

The mappings $\boldsymbol{C}$, $\boldsymbol{D}$, and $\boldsymbol{K}$ are all parameterized by convolutional layers. This choice is motivated by the inherent spatial structure in the neural image data. Convolutional layers are well-suited for hierarchically capturing local spatial

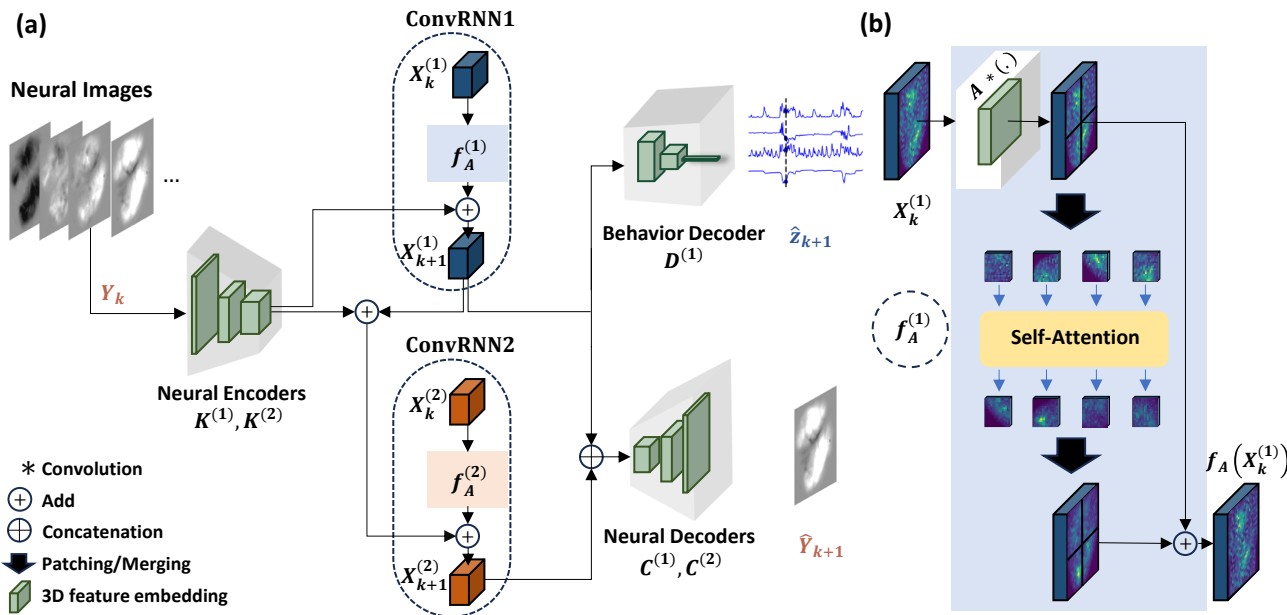

*Figure 1.* **(a)** A schematic representation of SBIND for jointly modeling neural image and behavior data. SBIND uses two ConvRNNs to learn behaviorally relevant neural dynamics (**ConvRNN1**) and behaviorally irrelevant neural dynamics (**ConvRNN2**) by separating the latent states into two subsets, $\mathbf{X}_k^{(1)}$ and $\mathbf{X}_k^{(2)}$, respectively. **(b)** The recurrence function, $\boldsymbol{f}_A^{(1)}(.)$ in **ConvRNN1** captures both spatial and temporal dependencies by applying a convolutional layer that captures local information followed by self-attention on patches of the latent state images to learn global spatial information. $\boldsymbol{f}_A^{(2)}(.)$ applies a similar function to $\mathbf{X}_k^{(2)}$ using a different set of parameters. The neural encoders are shallow convolutional networks designed to locally process the input images and downsample them into lower-dimensional latent representations. The behavior decoder predicts the behavior time-series based on the behaviorally relevant latent $\mathbf{X}_{k+1}^{(1)}$, while the neural decoder uses both the behaviorally relevant and irrelevant latent states, $\mathbf{X}_{k+1}^{(1)}$ and $\mathbf{X}_{k+1}^{(2)}$, to predict the neural image time-series.

dependencies and features in such data (LeCun et al., 1998). Specifically, $K$ utilizes a few nonlinear convolutional layers to downsample the neural images and extract local features. The neural decoder $C$ uses convolutional layers to upsample and transform the latent representation back to the original image space to predict the neural images one-step into the future. For $D$, convolutional layers decode behavior from the latent representation, with additional fully connected layers incorporated to decode continuous behavioral data that may not have a spatial structure.

The recursion function, $\boldsymbol{f}_A(\cdot)$, is formulated as $\boldsymbol{f}_A(.) = \text{GlobalAttn}(\boldsymbol{A}*(.))$, where $\text{GlobalAttn}$ represents the self-attention mechanism, $*$ indicates the convolution operator, and $\boldsymbol{A}$ is a set of convolutional kernels. $\boldsymbol{f}_A(\cdot)$ is designed to capture spatiotemporal dependencies in the latent state dynamics not only locally but also globally. To do so, it utilizes a convolutional layer, $\boldsymbol{A}$, to aggregate local features in the latent states. To additionally capture global context, a self-attention mechanism is incorporated (Vaswani et al., 2017). Within each time step, we divide the latent state image, $\mathbf{X}_k$, into patches, $\{\mathbf{x}_{k,1}, \mathbf{x}_{k,2}, ..., \mathbf{x}_{k,M}\}$, where each

patch $\mathbf{x}_{k,i}$ has dimensions $n_x \times P \times P$ and approximately represents features from a specific region of the brain (Figure 1b). Multi-head self-attention is then applied across these patches, allowing the model to learn spatial relationships between different regions (See details in Appendix A.1.2). Thus, as part of $\boldsymbol{f}_A$, the self-attention mechanism calculates spatial dependencies within each time step on the latent images, while the temporal dependencies are handled by the recurrent application of the entire $\boldsymbol{f}_A$ function (Equation 2). This combination of convolutional and self-attention layers constitutes the recursion function, $\boldsymbol{f}_A(\mathbf{X}_k)$, which is summed with the encoded neural image from the current step, $\boldsymbol{K}(\mathbf{Y}_k)$, to obtain $\mathbf{X}_{k+1}$. This enables the model to learn temporal dynamics as well as both local and global spatial patterns in neural image time-series data.

### 3.2. Neural and Behavioral Loss Functions and Distributions

We use one-step-ahead behavior decoding and neural image prediction losses to fit the parameters of the model. For neural prediction, we use a combination of L1, L2, and gradient

difference loss (GDL) (Mathieu et al., 2016). The GDL loss encourages the preservation of local image structure and improves the accuracy and structural fidelity of the predicted neural images (See formulation in Appendix A.1.3).

SBIND is designed to handle various behavioral data types, including continuous, categorical, and intermittently recorded behavior. This is achieved by modifying the training process and behavior loss. For Gaussian and categorical distributions, mean squared error (MSE) and cross-entropy losses are used, respectively, across time points. Moreover, in cases of intermittently recorded behavior, where behavior observations are sparse or missing, during training, our model utilizes a masked behavior loss to account for the missing observations while learning the latent representations from neural images as usual. Note, during inference, only neural images are used for inference of latents and decoding of behavior. Further details regarding the specific loss functions employed for each scenario are provided in Appendix A.1.3.

### 3.3. Dynamical Model Architecture and Learning

To effectively disentangle behaviorally relevant neural dynamics from other neural dynamics, we construct a model architecture consisting of two ConvRNNs, each integrated with self-attention mechanisms. To learn the model, we dedicate the latents of ConvRNN1 to capturing the behaviorally relevant dynamics to optimize behavior decoding and the latents of ConvRNN2 to finding the other neural dynamics to optimize neural prediction. We learn these two parts of our model sequentially for simpler interpretation and separation of the latents (Sani et al., 2024), though it is straightforward to learn them simultaneously with a combined neural-behavioral loss.

**Behaviorally Relevant Dynamics.** The first ConvRNN integrated with self-attention mechanisms focuses on finding the behaviorally relevant latents, decoding behavior, and capturing the corresponding neural dynamics. This ConvRNN is parameterized by $\boldsymbol{f}_A^{(1)}(\cdot)$, $\boldsymbol{K}^{(1)}(\cdot)$, $\boldsymbol{C}^{(1)}(\cdot)$, and $\boldsymbol{D}^{(1)}(\cdot)$. In the first phase of learning, $\boldsymbol{f}_A^{(1)}(\cdot)$, $\boldsymbol{K}^{(1)}(\cdot)$, and $\boldsymbol{D}^{(1)}(\cdot)$ are optimized to minimize the error in predicting behavior from the neural activity. This allows the model to learn a latent state representation, denoted as $\mathbf{X}_k^{(1)}$, that captures the behaviorally relevant neural dynamics. Also, $\boldsymbol{C}^{(1)}(\cdot)$ is optimized to reconstruct the neural images from the learned behaviorally relevant states, $\mathbf{X}_k^{(1)}$.

**Other Neural Dynamics.** The second ConvRNN integrated with self-attention mechanisms focuses on learning the remaining neural dynamics that are not captured by the first ConvRNN. This ConvRNN is parameterized by $\boldsymbol{f}_A^{(2)}(\cdot)$, $\boldsymbol{K}^{(2)}(\cdot)$, and $\boldsymbol{C}^{(2)}(\cdot)$. It takes as input the neural images, as well as the behaviorally relevant states from the first

ConvRNN, $\mathbf{X}_k^{(1)}$, as fixed values. In the second phase of learning, $\boldsymbol{f}_A^{(2)}(\cdot)$, $\boldsymbol{K}^{(2)}(\cdot)$, and $\boldsymbol{C}^{(2)}(\cdot)$ are optimized to minimize the error in predicting the neural image dynamics from both $\mathbf{X}_k^{(1)}$ and $\mathbf{X}_k^{(2)}$. Doing so allows $\mathbf{X}_k^{(2)}$ to capture neural image patterns not already captured by $\mathbf{X}_k^{(1)}$. This process cleanly separates the behaviorally relevant latents $\mathbf{X}_k^{(1)}$ from other latents $\mathbf{X}_k^{(2)}$.

The full inference model and learning details are formulated in Appendix A.1.1 and Appendix A.1.4, respectively.

### 3.4. Metrics and Evaluation

After training the model, we compute the one-step-ahead predictions of neural images and behavior on the test set using Equation 2 based only on neural image data. We use 5-fold cross-validation for widefield calcium imaging datasets and 10-fold cross-validation for each fUSI session. We report several metrics depending on the nature of the task. For neural prediction, we compute the MSE and coefficient of determination ($R^2$) between the predicted and observed neural images for one-step-ahead prediction. For behavior decoding, we use different metrics based on the type of behavioral data. We compute the MSE for one-step-ahead decoding of continuous behavioral data. For categorical data, we report accuracy and Area Under the Curve (AUC), and for imbalanced datasets, we use the F1-score.

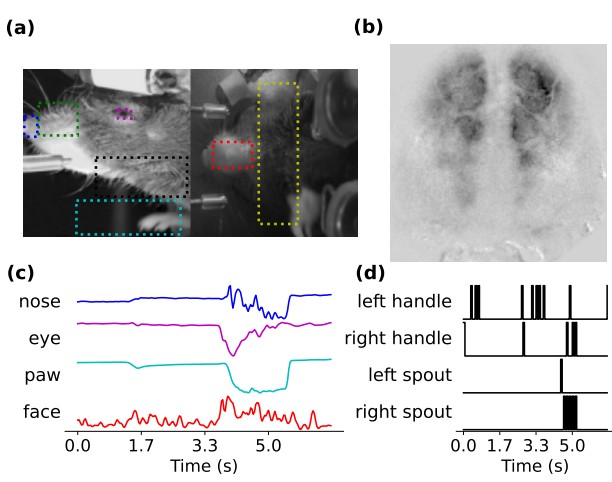

*Figure 2.* **(a)** Behavior videos recorded from a head-fixed mouse reporting the spatial position of visual or auditory stimuli. **(b)** Example widefield neural image. **(c)** Extracted continuous behavior from ROIs in behavior videos for dataset WFCI 1. **(d)** Behavior as 4 binary traces for dataset WFCI 2.

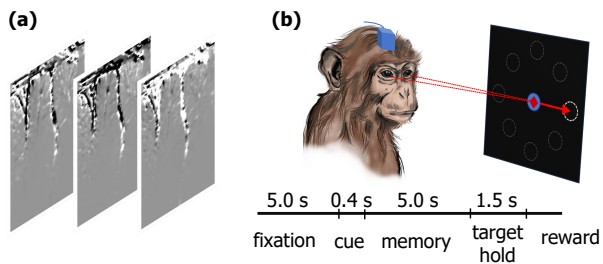

*Figure 3.* **(a)** samples frames of fUSI dataset. **(b)** Memory-guided saccade task timeline. The monkey fixated, viewed a brief peripheral cue (2 or 8 locations), maintained fixation during a memory period, and then made a saccade to the remembered target location.

# 4. Experiments

## 4.1. Experimental Details

### 4.1.1. DATASETS

We evaluate our model on three different neural-behavioral datasets: two publicly-available widefield calcium imaging (WFCI) datasets (Churchland et al., 2019) (Figure 2) and one publicly-available fUSI dataset (Griggs et al., 2023) (Figure 3).

**WFCI 1:** In this dataset, neural activity across the mouse dorsal cortex was optically recorded (Figure 2b), preprocessed, and downsampled to 1x128x128 pixel images. This dataset consists of 248 trials (49008 neural image frames) in which mice indicated the perceived spatial location (left or right) of an auditory or visual stimulus by licking the corresponding spout. Concurrently, behavior videos were recorded using two cameras (Figure 2a). We extracted 14 dimensions of continuous behavior from ROIs in the videos (Figure 2c). This continuous behavioral data was used for behavior decoding (see Section A.4.1 for details).

**WFCI 2:** This dataset comprises 412 trials (38927 neural image frames) with the same trial structure and neural recordings as WFCI 1. However, instead of behavior videos, four binary sensor traces were recorded from handles and spouts, providing categorical behavioral data for decoding (Figure 2d).

**fUSI:** This dataset consists of 13 sessions of functional ultrasound imaging recordings from a non-human primate performing memory-guided saccade or reach movements to one of 2 or 8 peripheral targets (Figure 3b). Functional ultrasound images were recorded at 2 Hz (Figure 3a). Behavior for this dataset consists of the directions the monkey saccaded to in successful trials. Thus, we use a categorical distribution for the behavior. Also, in this case, since there is one target per trial, we take behavior as available only during the period when the monkey was actually fixating on

the target and as missing during other periods of the trial. This provides an intermittently recorded behavior type for validation. Each session has $154.77 \pm 93.75$ successful trials, and each trial has a length of 15 seconds, corresponding to 30 image frames (see Appendix A.4.2 for more details).

### 4.1.2. IMPLEMENTATION DETAILS

We used three convolutional layers for both the encoder, $K$, and the decoder, $C$, to capture local spatiotemporal features and downsample the feature maps in the image dimension to 32x32. The transition function, $A$, is parameterized by a single convolutional layer, and $f(\cdot)$ is parameterized by multi-head self-attention (Vaswani et al., 2017) with 8 heads and an embedding dimension of 256, applied to patches of size $n_1 \times 4 \times 4$ (ConvRNN1) and $n_2 \times 4 \times 4$ (ConvRNN2). For behavior decoding, $D$ uses a few convolutional layers followed by a fully connected layer. A single inference step of the SBIND model takes 17.9 ms on average on an NVIDIA RTX 6000 Ada Generation GPU, which is shorter than the effective sampling rate of up to 10 Hz used in fUSI (Rabut et al., 2024; Macé et al., 2011) and sampling rate of 30 Hz used in WFCI, suggesting potential feasibility for real-time applications such as BCIs. See Appendix A.1.6 and Table A.1 for details on training information, hyperparameter tuning, and the choice of hyperparmeters for each of the datasets.

## 4.2. All SBIND Model Components Contribute to Accurate Neural Image Modeling

Here, we perform ablation studies by systematically removing or modifying specific parts of the model and comparing the resulting performances using relevant behavior decoding and neural prediction metrics (Table 1). We find that each component in our SBIND contributes to accurate neural-behavioral prediction as follows.

First, we assess the importance of using convolutional layers compared to Multi-Layer Perceptrons (MLPs) by constructing *MLP-SBIND*. *MLP-SBIND* parameterizes all the mappings with MLPs, with an option to also incorporate commonly used preprocessing methods for WFCI data. We show that even using this preprocessing – whether LocaNMF or PCA – as is done in current neural image models, this approach underperforms SBIND in both neural and behavior predictions due to lack of inductive bias for image data, unlike SBIND, which has convolutional layers as a component (Table 1).

Next, we assess the importance of our disentangled model architecture and neural-behavioral losses during learning. We construct *SBIND-Unsup* to train only the second phase of our model, i.e., learning neural dynamics in an unsupervised manner, which leads to just a single set of latents. We find that this ablated model exhibits inferior behavior de-

*Table 1.* One-step-ahead behavior decoding and neural prediction performances (Mean ± SEM) for various ablations of SBIND across 5 folds for widefield (WFCI) datasets in terms of mean-squared error (MSE) and/or F1-score as appropriate. As indicated by the arrows, lower is better for MSE and higher is better for F1-score. See Tables A.1 and A.2 for additional comparisons.

| | | WFCI 1 | | WFCI 2 | |
|---|---|---|---|---|---|
| MODEL | PREPROCESSING | BEH. MSE ↓ | NEUR. MSE ↓ | BEH. F1-SCORE ↑ | NEUR. MSE ↓ |
| MLP-SBIND | FLATTEN | 0.6383 ± 0.0281 | 0.1239 ± 0.0040 | 0.3066 ± 0.0108 | 0.1889 ± 0.0052 |
| MLP-SBIND | LOCANMF | 0.5980 ± 0.0253 | 0.0539 ± 0.0016 | 0.3804 ± 0.0072 | 0.2467 ± 0.0106 |
| MLP-SBIND | PCA | 0.6067 ± 0.0245 | 0.0589 ± 0.0029 | 0.2998 ± 0.0289 | 0.2834 ± 0.0241 |
| SBIND-UNSUP | - | 0.5413 ± 0.0185 | **0.0403 ± 0.0020** | 0.3985 ± 0.0194 | **0.1497 ± 0.0116** |
| SBIND NOATT | - | 0.5392 ± 0.0203 | 0.0552 ± 0.0013 | 0.4039 ± 0.0191 | 0.1948 ± 0.0047 |
| SBIND W/O $f_A$ | - | 0.7866 ± 0.0447 | 0.0812 ± 0.0033 | 0.3645 ± 0.0197 | 0.2150 ± 0.0032 |
| SBIND | - | **0.4955 ± 0.0254** | **0.0414 ± 0.0029** | **0.4569 ± 0.0036** | **0.1644 ± 0.0090** |

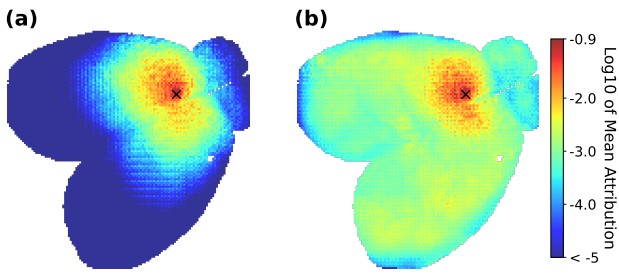

*Figure 4.* Mean contribution of all brain regions to predicting the activity of the pixel marked by × in the brain map, using **(a)** SBIND NoAtt vs. **(b)** SBIND. See Figure A.4 for more examples.

coding compared to SBIND with the same number of latent states (Table 1), highlighting the importance of our method for disentangling behaviorally relevant dynamics in neural image data.

Next, we investigate the recurrence unit by constructing *SBIND w/o $f_A$* that removes the recurrence (see Equation A.13). *SBIND w/o $f_A$* fails to capture long-term temporal dependencies, leading to inferior neural-behavioral predictions than SBIND (Table 1). Finally, to show the importance of the self-attention mechanism, we build *SBIND NoAtt* that removes the self-attention layer in the recurrence mapping, simplifying the recurrence function to a local convolutional layer. *SBIND NoAtt* has inferior performance in both behavioral and neural predictions (Table 1), showing the importance of self-attention for capturing global dependencies. To further explore this result, we increased the patch size used in the self-attention mechanism and observed improved neural prediction performance (Figure A.6). This suggests that larger patches allow the model to focus on more global information, complementing the local processing of the convolutional layers.

To pinpoint the brain regions that drive the neural predic-

tions, we employed Integrated Gradient implementation in Captum framework (Sundararajan et al., 2017; Kokhlikyan et al., 2020) to assess the contribution of brain-wide activity to the prediction of a specific target location in the brain. We averaged the attribution of all brain regions in predicting the target location across all frames of the test set in WFCI 1 data. The resulting attribution maps (Figure 4) demonstrate that SBIND, unlike SBIND NoAtt, leverages more global information for neural prediction. This confirms that the self-attention mechanism enables the model to capture dependencies beyond the local receptive fields of the convolutional layers, highlighting the importance of self-attention in understanding whole-brain dynamics.

### 4.3. Comparison with Existing Neural-Behavioral Models

We also find that SBIND outperforms recent neural-behavioral models on the same two WFCI datasets as in the previous section (Table 2). First, we compared SBIND with DPAD (Sani et al., 2024), which has nonlinearity options in its RNN and its encoder-decoder architecture. Despite this flexibility, DPAD performs worse in behavior decoding and neural prediction for neural image modalities, whether taking flattened widefield images as input or using preprocessing on the images in the form of LocaNMF or PCA (Table 2). This highlights the benefits of incorporating spatial priors for neural images in the model architecture, as in SBIND. Using common preprocessing techniques improves DPAD's neural prediction but does not change its behavior decoding performance, both of which still remain inferior to SBIND.

Next, we find that SBIND outperforms CEBRA (Schneider et al., 2023) in neural-behavioral prediction, regardless of whether CEBRA uses common preprocessing techniques, such as LocaNMF or PCA, or not (Table 2). CEBRA's low neural prediction performance (Table 2) can be attributed to the fact that it does not learn any residual dynamics for

*Table 2.* Behavior decoding MSE or F1-score and neural prediction MSE (Mean $\pm$ SEM) across folds for each dataset. As indicated by the arrows, lower is better for MSE and higher is better for F1-score. More detailed comparisons are provided in Table A.3 and Table A.4.

| Model | Preprocessing | WFCI 1 | | WFCI 2 | |
|---|---|---|---|---|---|
| | | Beh. MSE $\downarrow$ | Neur. MSE $\downarrow$ | Beh. F1-score $\uparrow$ | Neur. MSE $\downarrow$ |
| DPAD | LocaNMF | $0.5877 \pm 0.0226$ | $0.0543 \pm 0.0009$ | $0.3613 \pm 0.0145$ | $0.2483 \pm 0.0092$ |
| DPAD | PCA | $0.6164 \pm 0.0254$ | $0.0628 \pm 0.0030$ | $0.2688 \pm 0.0299$ | $0.2550 \pm 0.0063$ |
| DPAD | Flatten | $0.6179 \pm 0.0270$ | $0.1302 \pm 0.0013$ | $0.3177 \pm 0.0091$ | $0.2211 \pm 0.0071$ |
| CEBRA | LocaNMF | $0.6250 \pm 0.0194$ | $0.4976 \pm 0.0241$ | $0.3113 \pm 0.0183$ | $0.4363 \pm 0.0168$ |
| CEBRA | PCA | $0.6312 \pm 0.0239$ | $0.6856 \pm 0.0222$ | $0.3005 \pm 0.0235$ | $0.5323 \pm 0.0245$ |
| CEBRA | Flatten | $0.5995 \pm 0.0228$ | $0.7032 \pm 0.0201$ | $0.2909 \pm 0.0116$ | $0.2795 \pm 0.0109$ |
| SBIND | - | $\mathbf{0.4955 \pm 0.0254}$ | $\mathbf{0.0414 \pm 0.0029}$ | $\mathbf{0.4569 \pm 0.0036}$ | $\mathbf{0.1644 \pm 0.0090}$ |

neural prediction and relies on the embeddings guided by behavior. Figures A.2 illustrates that CEBRA primarily captures neural activity related to the observed behavior, with limited ability to predict activity in other brain regions. This highlights that neural activity encodes more information than just the studied behavior (Musall et al., 2019), and it is important to model residual neural dynamics to gain a more complete understanding of brain function.

Finally, in two additional baseline comparisons, SBIND again demonstrated its superiority for joint neural image and behavioral modeling by outperforming adapted versions of STNDT (Le & Shlizerman, 2022) and TNDM (Hurwitz et al., 2021). Originally developed for spiking electrophysiology data, we adapted these methods for our widefield imaging data as follows: we used LocaNMF features as input, placed a Gaussian prior over these inputs, and trained the models using an MSE loss instead of their original Poisson likelihood. Even with these adaptations, STNDT, which employs two separate sets of Transformers to capture spatial and temporal information, still underperforms in both neural and behavioral predictions compared to SBIND (Table A.5). Similarly, SBIND surpassed the adapted TNDM, a sequential variational autoencoder that learns two sets of latent factors for behaviorally relevant and irrelevant dynamics using a mixed neural-behavioral objective. These results consistently highlight the strength of SBIND's model design in modeling raw spatiotemporal neural images and disentangling behaviorally relevant neural dynamics.

### 4.4. Functional Ultrasound Imaging Data Results

For the fUSI dataset, behavior consisted of the target the monkey reached or saccaded to for each trial; thus, behavior was only taken as available for the time-steps in the trial during which the monkey was fixated on the target. This gave us a categorical and intermittently recorded behavior time-series for modeling. In the 2-directional tasks, we used binary target classification with a binary cross-entropy loss. In 8-directional tasks, similar to (Griggs et al., 2024),

we used a multi-decoder approach in the decoder mapping, $D^{(1)}$, to predict the vertical and horizontal directions (left-right-stationary) (Appendix A.4.2). During evaluation, we used the latent state at the last time-step in the trial to predict the direction of movement. We first performed ablation studies on all sessions. The ablated models underperformed SBIND in neural-behavioral prediction, again showing the importance of each component in our model (Table A.7 and Table A.8).

We then compared SBIND with other deep learning baselines as well as the original method used in (Griggs et al., 2024) for this fUSI dataset – i.e., PCA + linear discriminant analysis (LDA). For DPAD, we made similar decoder modifications to classify both 2-directional and 8-directional movements. For CEBRA, we found that labeling the entire trial with the target location yielded better embeddings for the decoding task than using only the samples during target fixation (See Appendix A.4.2 for details). As shown in Table 3, all models are inferior to SBIND in behavior decoding and neural prediction, again showing the importance of capturing image spatiotemporal structures through our ConvRNN and self-attention mechanisms.

## 5. Discussion

We develop SBIND, a novel approach for modeling dynamics in neural image data and their relationship to behavior, and demonstrate its success for two diverse neural imaging modalities: optical widefield and focused ultrasound imaging. We address the challenges of high-dimensionality and complex spatiotemporal patterns in these image modalities by designing a convolutional recurrent architecture combined with a self-attention mechanism. This enables us to learn both local and long-range spatiotemporal dynamics directly from raw neural image data without the need for preprocessing. Additionally, we tackle the challenge of prevalent behaviorally irrelevant dynamics in these images by disentangling the behavior-predictive dynamics while

*Table 3.* Behavior decoding accuracy (quantified as proportion of trials whose target was correctly decoded) and neural prediction MSE (Mean $\pm$ SEM) across 10 folds and all sessions in the fUSI dataset. As indicated by the arrows, lower is better for MSE and higher is better for accuracy. See Table A.7 and Table A.8 for comparison with ablated variants of SBIND. See Table A.9 for more comparisons.

| | | 2-DIRECTIONAL SESSIONS | | 8-DIRECTIONAL SESSIONS | |
|---|---|---|---|---|---|
| MODEL | PREPROCESSING | BEH. ACCURACY ↑ | NEUR. MSE ↓ | BEH. ACCURACY ↑ | NEUR. MSE ↓ |
| LDA | PCA | $0.7001 \pm 0.0177$ | - | $0.2806 \pm 0.0204$ | - |
| DPAD | FLATTEN | $0.5631 \pm 0.0191$ | $0.8409 \pm 0.0072$ | $0.1840 \pm 0.0131$ | $0.9236 \pm 0.0083$ |
| DPAD | PCA | $0.6783 \pm 0.0167$ | $0.6391 \pm 0.0056$ | $0.2864 \pm 0.0183$ | $0.7109 \pm 0.0034$ |
| CEBRA | FLATTEN | $0.6813 \pm 0.0193$ | $1.6647 \pm 0.0128$ | $0.2705 \pm 0.0165$ | $1.6966 \pm 0.0144$ |
| CEBRA | PCA | $0.7276 \pm 0.0203$ | $1.5832 \pm 0.0088$ | $0.2633 \pm 0.0163$ | $1.6934 \pm 0.0190$ |
| SBIND | - | $\mathbf{0.7300 \pm 0.0191}$ | $\mathbf{0.4725 \pm 0.0165}$ | $\mathbf{0.3521 \pm 0.0201}$ | $\mathbf{0.3919 \pm 0.0107}$ |

simultaneously capturing other ongoing neural processes. In contrast to other neural-behavioral models that rely on preprocessing, SBIND assumes an image prior on the input, enabling the learning of spatiotemporal information for modeling neural images and predicting behavior. Our model outperforms existing neural-behavioral models in predicting behavior, regardless of the prior distribution of the behavior of interest. Furthermore, to our knowledge, our work presents the first dynamical latent state model for fUSI data.

In this study, we focused on neural imaging modalities with relatively high temporal resolution, namely widefield calcium imaging and functional ultrasound imaging that also has potential for BCIs. This is in contrast to some imaging modalities such as functional magnetic resonance imaging (fMRI) that have lower temporal resolution and are largely not usable for portable BCIs. Indeed, for this reason, prior deep learning approaches for fMRI are different in goals compared to our work. These fMRI models largely focus on classifying task conditions or participant demographics (e.g., age, sex) (Gadgil et al., 2020; Kan et al., 2022; Malkiel et al., 2022; Li et al., 2023) rather than modeling behaviorally relevant temporal dynamics and disentangling them from other neural temporal dynamics, which is our goal here.

Our work focused on learning the model using data from a single session or animal. An important direction for future work is to extend SBIND to enable multi-session learning, for example by adding session specific read-in and read-out layers while keeping the model core uniform across sessions. Doing so may both improve performance and allow for investigating how whole-brain local and global spatiotemporal dynamics change across sessions and animals using SBIND. Indeed, for training their PCA-LDA decoder on fUSI data, Griggs et al. (2024) showed that using multi-session data for pretraining the decoder can be helpful since day-to-day recordings have similarities in their neural representations. Finally, tracking time-varying dynamics through adaptive learning can be another interesting future direction (Ahmadipour et al., 2021; Yang et al., 2021).

SBIND offers significant practical advantages for BCIs (Shanechi, 2019; Shenoy & Carmena, 2014; Oganesian & Shanechi, 2024). SBIND's inference time is faster than typical sampling rates of WFCI and fUSI (Rabut et al., 2024; Macé et al., 2011). This property, coupled with SBIND's recursive inference, can enable real-time and computationally efficient modeling of neural images and behavior decoding. Also, SBIND's recurrent architecture inherently supports neural forecasting of behavior several steps into the future, without a need for model retraining. These capabilities, combined with SBIND's successful demonstration on diverse imaging modalities, could help enable future non-invasive BCIs using fUSI, which is a recent promising modality whose dynamical modeling was previously unexplored.

## Impact Statement

This work can enable the decoding of non-invasive fUSI neural modalities, which may have positive societal impact by facilitating the design of non-invasive BCIs that restore lost movement in paralyzed patients. Other than this, this paper also presents work whose goal is to advance the field of Machine Learning. In this regard, there are many potential societal consequences of our work, none which we feel must be specifically highlighted here.

## Acknowledgment

This work was supported, in part, by National Institutes of Health (NIH) grants R01MH123770, R61MH135407, and RF1DA056402, NIH Director's New Innovator Award DP2-MH126378, and National Science Foundation (NSF) CR-CNS program award IIS-2113271. We thank Simon Musall and Anne Churchland for making the widefield imaging datasets publicly available and providing guidance on their usage. We also thank the Andersen and Shapiro Labs for making the fUSI dataset publicly available. Finally, we thank Parsa Vahidi for his help with the manuscript and baseline comparisons.

## Reproducibility Statement

To ensure the reproducibility of our work, we are sharing the code for SBIND at https://github.com/ShanechiLab/SBIND/. We also provide model hyperparameters and training details in Appendix A.1.6. The Widefield Calcium (Churchland et al., 2019) and functional Ultrasound Imaging (Griggs et al., 2023) datasets used in our main experiments are publicly available; details of the preprocessing applied to these datasets are provided in Appendix A.4, and our preprocessing framework for these datasets is also shared in our repository for those interested in reproducing the main experiments presented in this work.

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

# A. Appendix

## A.1. Methods Supplementary Details

### A.1.1. FORMULATION OF THE FULL INFERENCE MODEL

Our model employs a two-RNN architecture to effectively capture and disentangle behaviorally relevant neural dynamics from other neural dynamics. The first RNN (*ConvRNN1*) focuses on behaviorally relevant dynamics before the second RNN (*ConvRNN2*) learns the remaining dynamics.

*ConvRNN1* is parameterized by $\boldsymbol{f}_A^{(1)}(\cdot)$, $\boldsymbol{K}^{(1)}(\cdot)$, $\boldsymbol{C}^{(1)}(\cdot)$, and $\boldsymbol{D}^{(1)}(\cdot)$. It takes as input the neural images, $\mathbf{Y}_k$, and outputs a latent state representation, $\mathbf{X}_k^{(1)} \in \mathbb{R}^{n_1 \times H' \times W'}$, that captures the behaviorally relevant neural dynamics.

*ConvRNN2* is parameterized by $\boldsymbol{f}_A^{(2)}(\cdot)$, $\boldsymbol{K}^{(2)}(\cdot)$, and $\boldsymbol{C}^{(2)}(\cdot)$. It takes as input the neural images, $\mathbf{Y}_k$, as well as the latent states from *ConvRNN1*, $\mathbf{X}_k^{(1)}$, and outputs a latent state representation, $\mathbf{X}_k^{(2)} \in \mathbb{R}^{n_2 \times H' \times W'}$, that captures the residual neural dynamics not captured by *ConvRNN1*.

In Equation 2, both latent states were combined for simplicity. The full inference model (Figure 1a) can be formulated as follows:

$$\begin{cases} \mathbf{X}_{k+1}^{(1)} &= \boldsymbol{f}_A^{(1)}(\mathbf{X}_k^{(1)}) + \boldsymbol{K}^{(1)}(\mathbf{Y}_k) \\ \mathbf{X}_{k+1}^{(2)} &= \boldsymbol{f}_A^{(2)}(\mathbf{X}_k^{(2)}) + \boldsymbol{K}^{(2)}(\mathbf{Y}_k, \mathbf{X}_{k+1}^{(1)}) \\ \hat{\mathbf{Y}}_k &= \boldsymbol{C}^{(1)}(\mathbf{X}_k^{(1)}) + \boldsymbol{C}^{(2)}(\mathbf{X}_k^{(2)}) \\ \hat{\mathbf{z}}_k &= \boldsymbol{D}^{(1)}(\mathbf{X}_k^{(1)}) \end{cases} \tag{A.1}$$

where $\hat{\mathbf{Y}}_k \in \mathbb{R}^{n_y \times H \times W}$ is the predicted neural images and $\hat{\mathbf{z}}_k \in \mathbb{R}^{n_z}$ is the predicted behavior at time index, $k$.

The full set of states can be denoted as $\mathbf{X}_k \in \mathbb{R}^{n_x \times H' \times W'}$ where $n_x = n_1 + n_2$, and is achieved by concatenating the image latent states in the channel dimensions. As seen in Equation A.1, $\mathbf{X}_k^{(1)}$ is calculated independently of $\mathbf{X}_k^{(2)}$. As discussed in Section A.1.4, $\mathbf{X}_k^{(1)}$ are learned to decode behavior, so they essentially capture behaviorally relevant neural dynamics. $\mathbf{X}_k^{(2)}$ learn other neural dynamics by optimizing for neural prediction. This is achieved by passing the neural images and states from *ConvRNN1* as residuals in the calculation of the second set of states. The above formulation can be written in combined form as in Equation 2, where:

$$\mathbf{X}_k = \begin{bmatrix} \mathbf{X}_k^{(1)} & \mathbf{X}_k^{(2)} \end{bmatrix}^T,$$

$$\boldsymbol{f}_A(\mathbf{X}_k) = \begin{bmatrix} \boldsymbol{f}_A^{(1)}(\mathbf{X}_k^{(1)}) \\ \boldsymbol{f}_A^{(2)}(\mathbf{X}_k^{(2)}) \end{bmatrix} = \begin{bmatrix} \mathrm{GlobalAttn}^{(1)}(\boldsymbol{A}^{(1)} * \mathbf{X}_k^{(1)}) \\ \mathrm{GlobalAttn}^{(2)}(\boldsymbol{A}^{(2)} * \mathbf{X}_k^{(2)}) \end{bmatrix},$$

$$\boldsymbol{K}(\mathbf{Y}_k) = \begin{bmatrix} \boldsymbol{K}^{(1)}(\mathbf{Y}_k) \\ \boldsymbol{K}^{(2)}(\mathbf{Y}_k, \mathbf{X}_{k+1}^{(1)}) \end{bmatrix},$$

$$\boldsymbol{C}(\mathbf{X}_k) = \boldsymbol{C}^{(1)}(\mathbf{X}_k^{(1)}) + \boldsymbol{C}^{(2)}(\mathbf{X}_k^{(2)}),$$

$$\boldsymbol{D}(\mathbf{X}_k) = \boldsymbol{D}^{(1)}(\mathbf{X}_k^{(1)})$$

These notations connect the full inference model (Equation A.1) and the combined form in Equation 2. Also, $\boldsymbol{f}_A(.) = \mathrm{GlobalAttn}(\boldsymbol{A} * (.))$, where $\mathrm{GlobalAttn}$ represents the self-attention mechanism, and $\boldsymbol{A}$ represents the convolutional kernels applied on the states prior to self-attention.

### A.1.2. DETAILS OF SELF-ATTENTION OPERATION

The recurrence function, $\boldsymbol{f}_A^{(1)}(\cdot)$, is designed to capture spatiotemporal dependencies in the latent state representation of *ConvRNN1* both locally and globally (Figure A.1). For simplicity, we explain details of applying the recurrence function $\boldsymbol{f}_A^{(1)}(\cdot)$, on the latent state at time index $k$, $\mathbf{X}_k^{(1)}$. However, the same function is applied at all other time indices recurrently.

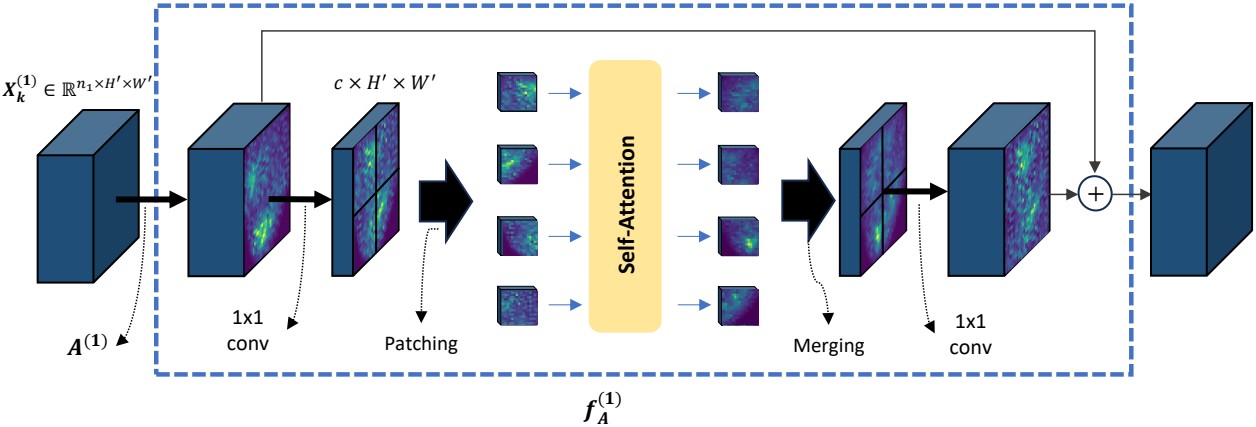

*Figure A.1.* Details of self-attention mechanism in $\boldsymbol{f}_A^{(1)}$. Self-attention is used here to capture global spatial relationships within the latent state. $\boldsymbol{f}_A^{(2)}$ applies the same self-attention mechanism to $\mathbf{X}_k^{(2)}$ using a different set of parameters

$\boldsymbol{A}^{(1)}$ is a single convolutional layer to capture local dependencies in the latent state. Additionally, the recurrence functions incorporate a self-attention mechanism to capture global context and long-range dependencies in the latent state dynamics. Here we explain the details of applying self-attention to $\mathbf{X}_k^{(1)}$.

**Reduction in channel dimension.** For $\boldsymbol{f}_A^{(1)}(\cdot)$, the process begins by passing $\mathbf{X}_k^{(1)}$ through a 1x1 convolutional layer to reduce the channel size from $n_x$ to $c$, resulting in a representation with dimensions $c \times H' \times W'$. This reduction in channel size is optional and makes subsequent self-attention computation more efficient. In our experiments, we found that reducing the channel dimension to $c = 2$ provided a good balance between computational and parameter efficiency, and model performance.

**Patching.** Next, the latent state representation is divided into patches, $\{\mathbf{x}_{k,1}^{(1)}, \mathbf{x}_{k,2}^{(1)}, ..., \mathbf{x}_{k,M}^{(1)}\}$, similar to the approach used in (Dosovitskiy et al., 2021). The total number of patches, $M$, is calculated as $M = (H' \times W')/(P \times P)$, where $P$ is the patch size. Each patch, $\mathbf{x}_{k,i}^{(1)}$, has dimensions $c \times P \times P$ and can be interpreted as a representation of features from roughly a specific brain region.

**Self-attention block.** The patches are then flattened into vectors of shape $cP^2$ and treated as tokens for the self-attention mechanism. A one-dimensional learnable embedding is added to each of the tokens (patches), so the self-attention layer is informed of the position of the embedding in the latent state images. Multi-head self-attention is applied to these tokens (Vaswani et al., 2017), allowing the model to learn spatial relationships between different patches. layer normalization is applied before and after the self-attention.

**Projecting back to the latent state representation.** After applying self-attention to the embedded patches, the patches are reshaped and rearranged into $c \times H' \times W'$ to get to the original spatial dimensions, effectively reversing the patching operation. A $1 \times 1$ convolutional layer then projects the $c \times H' \times W'$ latents to an $n_1 \times H' \times W'$ dimensional space. The resulting representation is added to the output of $\boldsymbol{A}^{(1)}$, forming the final output of the recurrence function.

This combination of convolutional layers and self-attention layers in the recurrence function enables the model to effectively capture both local and global spatial dependencies. The same recurrence function is applied to $\mathbf{X}_k^{(2)}$, which has dimensions $n_2 \times H' \times W'$, across all time points using a different set of parameters to learn a separate set of dynamics for the behaviorally irrelevant component of the neural images.

### A.1.3. LOSS FUNCTIONS

**Neural Prediction Loss** For neural prediction, we use a combination of L1, L2, and gradient difference loss (GDL) (Mathieu et al., 2016). The GDL loss encourages the preservation of local image structure by penalizing differences between

the gradients of the predicted and ground-truth images. This combined loss function aims to improve the accuracy and structural fidelity of the predicted neural images. The L1, L2, and GDL functions are defined as follows:

$$\mathcal{L}_{L1}(\hat{Y}_k, Y_k) = \sum_{i,j} |\hat{Y}_k^{i,j} - Y_k^{i,j}|, \tag{A.2}$$

$$\mathcal{L}_{L2}(\hat{Y}_k, Y_k) = \sum_{i,j} (\hat{Y}_k^{i,j} - Y_k^{i,j})^2, \tag{A.3}$$

$$\mathcal{L}_{grad}(\hat{Y}_k, Y_k) = \sum_{i,j} \left| ||Y_k^{i,j} - Y_k^{i-1,j}|| - ||\hat{Y}_k^{i,j} - \hat{Y}_k^{i-1,j}||| + |||Y_k^{i,j-1} - Y_k^{i,j}|| - ||\hat{Y}_k^{i,j-1} - \hat{Y}_k^{i,j}|||, \tag{A.4}$$

where $i, j$ index the spatial dimensions of the image. The total loss for neural image reconstruction is given by:

$$\mathcal{L}_{\mathbf{Y}} = \mathcal{L}_{L2}(\hat{Y}_k, Y_k) + \lambda_{L1}\mathcal{L}_{L1}(\hat{Y}_k, Y_k) + \lambda_{grad}\mathcal{L}_{grad}(\hat{Y}_k, Y_k), \tag{A.5}$$

where $\lambda_{L1}$ and $\lambda_{grad}$ are hyperparameters that control the relative weights of the L1 and GDL losses.

**Behavior Decoding Loss**  For behavior decoding, the loss function, $\mathcal{L}_{\mathbf{z}}$, is chosen based on the distribution and availability of data at each time point.

- **Continuous Behavior:** Assuming an isotropic Gaussian distribution for this type of behavior, we use the MSE loss.

- **Categorical Behavior:** We use class-weighted cross-entropy loss for categorically distributed behavior to address potential class imbalance (Lin et al., 2017). For instance, in the WFCI 2 dataset, where the mouse is not touching the sensors over 90% of the time, we assign a weight of 0.9 to class 1 (indicating when the mouse is touching a sensor—left handle, right handle, left spout, or right spout) and 0.1 to class 0 (indicating when the mouse is not touching a sensor).

- **Intermittently Recorded Behavior:** We utilize a masking strategy to handle intermittently recorded behavior. The behavior loss is calculated only at the sparse time points where the behavior is observed.

### A.1.4. TWO-PHASE LEARNING DETAILS

To disentangle behaviorally relevant neural dynamics from other neural dynamics, we design a model architecture with two ConvRNNs, each incorporating self-attention mechanisms. The parameters of these two ConvRNNs are learned in two sequential phases to achieve the disentanglement.

**Phase 1: Learning Behaviorally Relevant Dynamics**

First, the parameters of *ConvRNN1* - i.e., $\boldsymbol{f}_A^{(1)}(\cdot)$, $\boldsymbol{K}^{(1)}(\cdot)$, and $\boldsymbol{D}^{(1)}(\cdot)$ - and the behaviorally relevant latent states, $\mathbf{X}_k^{(1)}$, are learned to minimize the error in predicting behavior from the neural images. The following recurrent formula is used to predict behavior, $\hat{\mathbf{z}}_k^{(1)}$, one-step into the future:

$$\begin{cases} \mathbf{X}_{k+1}^{(1)} &= \boldsymbol{f}_A^{(1)}(\mathbf{X}_k^{(1)}) + \boldsymbol{K}^{(1)}(\mathbf{Y}_k) \\ \hat{\mathbf{z}}_k^{(1)} &= \boldsymbol{D}^{(1)}(\mathbf{X}_k^{(1)}) \end{cases} \tag{A.6}$$

The optimization is formulated as:

$$\min_{\boldsymbol{f}_A^{(1)}, \boldsymbol{K}^{(1)}, \boldsymbol{D}^{(1)}} \sum_k \mathcal{L}_{\mathbf{z}}(\mathbf{z}_k, \hat{\mathbf{z}}_k^{(1)}), \tag{A.7}$$

where $\mathcal{L}_{\mathbf{z}}$ is the loss function for behavior decoding chosen based on the distribution of behavior (Appendix A.1.3). $k \in [1, 2, ..., T]$ where $T$ is the total number of samples. This optimization ensures that *ConvRNN1* learns neural dynamics that are relevant to behavior.

Once *ConvRNN1* is learned, its parameters are fixed, and the decoder $C^{(1)}(\cdot)$ is optimized to predict neural images one step into the future from the learned latent states, $\mathbf{X}_k^{(1)}$. This can be formulated as:

$$\hat{\mathbf{Y}}_k^{(1)} = C^{(1)}(\mathbf{X}_k^{(1)}) \tag{A.8}$$

and the optimization is formulated as:

$$\min_{C^{(1)}} \sum_k \mathcal{L}_{\mathbf{Y}}(\mathbf{Y}_k, \hat{\mathbf{Y}}_k^{(1)}), \tag{A.9}$$

where $\mathcal{L}_{\mathbf{Y}}$ is the loss function for neural reconstruction, defined in Equation A.5.

**Phase 2: Learning Residual Neural Dynamics**

In this phase, the parameters of *ConvRNN2* - i.e., $f_A^{(2)}(\cdot)$, $K^{(2)}(\cdot)$, and $C^{(2)}(\cdot)$ - are learned to minimize the error in predicting the residual neural images, i.e., the part of neural images not predicted by the behaviorally relevant states of *ConvRNN1*. This is achieved by training *ConvRNN2* to predict the difference between the observed neural images and the neural images predicted by *ConvRNN1*. The residual neural predictions are calculated using the following recurrent formulation:

$$\begin{cases} \mathbf{X}_{k+1}^{(2)} &= f_A^{(2)}(\mathbf{X}_k^{(2)}) + K^{(2)}(\mathbf{Y}_k, \mathbf{X}_{k+1}^{(1)}) \\ \hat{\mathbf{Y}}_k^{(2)} &= C^{(2)}(\mathbf{X}_k^{(2)}) \end{cases} \tag{A.10}$$

This step learns the residual neural dynamics and the latent states, $\mathbf{X}_k^{(2)}$. The optimization is formulated as:

$$\min_{f_A^{(2)}, K^{(2)}, C^{(2)}} \sum_k \mathcal{L}_{\mathbf{Y}}(\mathbf{Y}_k - \hat{\mathbf{Y}}_k^{(1)}, \hat{\mathbf{Y}}_k^{(2)}) \tag{A.11}$$

This concludes learning the ConvRNNs and the total latent states $\mathbf{X}_k$. Note that in the two optimization steps in Equations A.9, and A.11, the optimization only controls and learns the parameters in the current optimization, and the parameters from previous optimizations are fixed.

A.1.5. ALTERNATIVE FORMULATION FOR THE INFERENCE MODEL

We can optionally concatenate the mappings from the encoder, $K(\mathbf{Y}_k)$, with the states of the current ConvRNN before feeding the latent states into the recurrence function. This can be formulated as:

$$\begin{cases} \mathbf{X}_{k+1}^{(1)} &= f_A^{(1)}(\mathbf{X}_k^{(1)}, K^{(1)}(\mathbf{Y}_k)) \\ \mathbf{X}_{k+1}^{(2)} &= f_A^{(2)}(\mathbf{X}_k^{(2)}, \mathbf{X}_{k+1}^{(1)}, K^{(2)}(\mathbf{Y}_k)) \\ \hat{\mathbf{Y}}_k &= C^{(1)}(\mathbf{X}_k^{(1)}) + C^{(2)}(\mathbf{X}_k^{(2)}) \\ \hat{\mathbf{z}}_k &= D^{(1)}(\mathbf{X}_k^{(1)}) \end{cases} \tag{A.12}$$

This can be thought of as including the information from the neural images at the current time index within the mapping. This does not change the dimensions of the latent states, but the input to the convolutional layer within the recurrence uses more kernels. This is a more general form of Equation A.1, and for the first two datasets, this form achieves slightly better performance (See Table A.6).

A.1.6. SBIND ARCHITECTURE AND IMPLEMENTATION DETAILS

The Neural Encoders, $K^{(1)}$ and $K^{(2)}$, each consist of three convolutional layers that downsample the input neural images to a $32 \times 32$ spatial resolution. These layers process the images statically and locally. To ensure stable learning, each convolutional layer is followed by batch normalization to normalize activations, and Leaky ReLU (Xu et al., 2015) is used as the activation function. Padding is applied to preserve spatial dimensions during convolutions.

The neural decoders, $\boldsymbol{C}^{(1)}$ and $\boldsymbol{C}^{(2)}$, consist of three transposed convolutional layers, which are designed to upsample the latent states and project them back into the neural image observation space. The same activation function and normalization are applied to these decoders.

The behavior decoder, $\boldsymbol{D}$, begins by further downsampling the $32 \times 32$ latent state to a $4 \times 4$ spatial resolution using three convolutional layers, each with stride 2, batch normalization, Leaky ReLU activation, and channel dropouts (Tompson et al., 2015). This reduction is followed by a fully connected layer to project the latent state into the behavior observation space.

The recurrence functions, $\boldsymbol{f}_A^{(1)}$ and $\boldsymbol{f}_A^{(2)}$, each use a single convolutional layer with $3 \times 3$ kernels to process the latent states locally. Afterward, the model applies multi-head self-attention to patches of the latent state images to capture long-range dependencies across the latent space. Each latent patch is mapped to a 256-dimensional embedding space, which is then used to compute the self-attention.

Training details, including the learning rate, optimizer choice, and hyperparameters of the mappings, are summarized in Table A.1.

**Hyperparameter Tuning:**

We use latent states with dimensions $n_x \times 32 \times 32$ across all experiments. The first $n_1$ channels of the latent states are used in *ConvRNN1* to model behaviorally relevant dynamics. Any additional channels are used for *ConvRNN2* to learn residual neural dynamics ($n_2 = \max(n_x - n_1, 0)$). We use $n_1 = 8$ across all experiments. The number of latent channels, $n_x$, was tuned in the set $\{1, 2, 4, 8, 16, 32\}$, where for $n_x \leq 8$ only *ConvRNN1* is trained, and for $n_x > 8$ *ConvRNN2* is learned in the second phase in addition to *ConvRNN1*. The patch size for the self-attention layer in $\boldsymbol{f}_A^{(1)}$ and $\boldsymbol{f}_A^{(2)}$ was tuned in the set $\{1, 2, 4, 8, 16\}$.

**Implementation Details :** For WFCI 1 dataset with 39200 neural image samples used for training, it takes 2445 seconds on average to run all 3 optimization steps for 80 epochs on an NVIDIA RTX 6000 Ada Generation GPU, and the inference takes 13.5 seconds on 9800 sequential samples.

### A.2. Ablation Details

#### A.2.1. MLP (PARAMETERIZED)-SBIND

This ablation explores the importance of convolutional layers in our model by replacing them with MLPs. It maintains the same two-phase learning scheme as SBIND, but uses MLPs to parameterize all the mappings in both phases and relies on MLPs to learn spatiotemporal structure in neural image data. The states $\mathbf{X}_k^{(1)}$ and $\mathbf{X}_k^{(2)}$ are vectors of shape $\mathbb{R}^{n_1 \times 1 \times 1}$ and $\mathbb{R}^{n_2 \times 1 \times 1}$, respectively, disregarding the spatial distribution in neural images.

Optionally, we use commonly used preprocessing techniques on widefield and ultrasound data to obtain a low-dimensional representation (as a vector) for the neural images. This representation is then used as input for training and inference on the model. After learning the model and obtaining predictions in the low-dimensional space, we project the prediction back to the neural image space to compare the performance in neural prediction with SBIND.

#### A.2.2. SBIND W/O $\boldsymbol{f}_A$

This ablation investigates the importance of recurrent neural networks in our model by removing the recurrence mapping, $\boldsymbol{f}_A(\cdot)$. This is essentially a Convolutional Autoencoder (CAE) that takes neural images as input and attempts to predict neural images one-step into the future. This is equivalent to our model without the recurrence function, just optimizing for $\boldsymbol{K}, \boldsymbol{C}$, and $\boldsymbol{D}$. This ablation is used to pinpoint the importance of using information from neural images more than one sample in the past for modeling. A behavior decoder mapping, $\boldsymbol{D}$ is trained downstream to project the latent representation of the CAE, $\mathbf{X}_k$, to the behavior of interest. The same loss functions from SBIND are used for neural and behavioral prediction. The inference of this model can be formulated as:

$$\begin{cases} \mathbf{X}_{k+1} &= \boldsymbol{K}(\mathbf{Y}_k) \\ \hat{\mathbf{Y}}_{k+1} &= \boldsymbol{C}(\mathbf{X}_{k+1}) \\ \hat{\mathbf{z}}_{k+1} &= \boldsymbol{D}(\mathbf{X}_{k+1}) \end{cases} \tag{A.13}$$

*Table A.1.* SBIND Model and Training Details and Hyperparameters Across All Datasets

| Component | Hyperparameter | WFCI 1 | WFCI 2 | fUSI |
|---|---|---|---|---|
| General | Input Dimensions | 1x128x128 | 1x128x128 | 1x128x128 |
| | $n_1$ | 8 | 8 | 8 |
| | $n_2$ | 8 | 24 | 24 |
| | Batch Size | 7 | 7 | 30 |
| | Sequence Length | 63 | 63 | 30 |
| | Learning Rate | 1e-3 | 1e-3 | 1e-3 |
| | LR Schedule | StepLR | StepLR | StepLR |
| | Max Training Epochs | 80 | 80 | 80 |
| | Weight Decay | 1e-6 | 1e-6 | 1e-6 |
| Neural Encoders $K^{(1)}$ ($K^{(2)}$) | Layers | 3 | 3 | 3 |
| | Kernel Size | 5x5 | 5x5 | 5x5 |
| | Strides | 2, 2, 1 | 2, 2, 1 | 2, 2, 1 |
| | Channel Dropout | 0 | 0 | 0 |
| | Num Kernels | 32, 32, $n_1$ ($n_2$) | 32, 32, $n_1$ ($n_2$) | 32, 32, $n_1$ ($n_2$) |
| Neural Decoder $C^{(1)}$($C^{(2)}$) | Layers | 2 | 2 | 2 |
| | Kernel Size | 5x5 | 5x5 | 5x5 |
| | Strides | 2 | 2 | 2 |
| | Channel Dropout | 0 | 0 | 0 |
| | Num Kernels | 32, 32, 1 | 32, 32, 1 | 32, 32, 1 |
| | $\lambda_{L1}$ | 2.0 | 2.0 | 2.0 |
| | $\lambda_{grad}$ | 0.3 | 0.3 | 0.3 |
| Behavior Decoder ($D^{(1)}$) | Conv Layers | 3 | 3 | 4 |
| | Kernel Size | 5x5 | 5x5 | 5x5 |
| | Strides | 2 | 2 | 2 |
| | Channel Dropout | 0.4 | 0.4 | 0.4 |
| | Num Kernels | 64, 64, 64 | 64, 64, 64 | 16, 16, 16, 16 |
| | FCN Layers | 1 | 1 | 1 |
| | FCN hidden units | 64 | 64 | 16 |
| Recurrence function $f_A^{(1)}$ ($f_A^{(2)}$) | Conv. Layers | 1 | 1 | 1 |
| | Kernel Size | 3x3 | 3x3 | 3x3 |
| | Strides | 1 | 1 | 1 |
| | Channel Dropout | 0 | 0 | 0 |
| | Hidden Dim | 48 | 48 | 48 |
| | Num Kernels | $n_1$ ($n_2$) | $n_1$ ($n_2$) | $n_1$ ($n_2$) |
| | Self-Attention Heads | 8 | 8 | 8 |
| | Patch Size | 8 | 8 | 8 |
| | Embedding Dim | 256 | 256 | 256 |
| | Positional Embedding | Learnable | Learnable | Learnable |
| | Num Patches | 16 | 16 | 16 |

### A.2.3. SBIND MSE $L_Y$

This ablation explores the effect of the neural loss function in Equation A.5 by removing the GDL and L1 loss components. This ablation uses the same model architecture and training procedure as SBIND, but trains the model using only the MSE loss for neural prediction in the optimizations of Equations A.9 and A.11.

### A.2.4. SBIND-UNSUP (UNSUPERVISED)

This ablation investigates the importance of disentangling behaviorally relevant dynamics by removing the first phase of our algorithm. This forces the model to learn all neural dynamics without considering their relevance to behavior, as it predicts neural data one step into the future without using behavior information. Effectively, this ablation sets $n_1 = 0$ and

$n_x = n_2$, as it learns $\mathbf{X}_k^{(2)}$ and *ConvRNN2* parameters while still using self-attention in the recurrence and convolutional layers. Because SBIND-Unsup only learns a single ConvRNN for neural prediction, it may learn behaviorally irrelevant neural dynamics in neural images, potentially resulting in inferior behavioral prediction performance.

We use the same latent state dimensions, $n_x \times H' \times W'$, whenever we compare the performance of SBIND with this variant.

### A.2.5. SBIND NoAtt

This ablation assesses the impact of the self-attention mechanism by removing it from the recurrence function, $\boldsymbol{f_A}(\cdot)$. This model still disentangles behaviorally relevant neural dynamics using the two-phase learning scheme. It also uses the same hyperparameters for all other mappings ($\boldsymbol{K}$, $\boldsymbol{C}$, and $\boldsymbol{D}$) and the same loss functions as SBIND. However, it does not utilize self-attention to capture long-range spatial information in the latent space, and consequently, in the neural image space. By removing self-attention, $\boldsymbol{f_A}(\cdot)$ simplifies to a local convolutional layer. This prevents the model from capturing dependencies between distant brain regions and using these dependencies for neural and behavioral prediction (Figure 4).

### A.3. Baseline Neural-behavioral Models and Preprocessing Methods

First, we list the two dimensionality reduction methods that are commonly used when working with widefield calcium and functional ultrasound imaging data. We use these as optional preprocessing steps to obtain a low-dimensional representation for neural-behavioral baselines. We compare our model with CEBRA (Schneider et al., 2023), which extracts latent embeddings informed by behavior and fits decoders for neural and behavior observations. Moreover, we compare our model performance with DPAD (Sani et al., 2024), which is a nonlinear dynamical model that learns behaviorally relevant neural dynamics.

### A.3.1. PCA

PCA is often performed on widefield calcium imaging data as a dimensionality reduction technique before performing modeling (Musall et al., 2019). Enough principal components are extracted to explain a sufficient amount of variance in the neural images. For functional ultrasound imaging, PCA serves the same purpose and is also employed to decode movement intentions (Griggs et al., 2024; Norman et al., 2021).

When using PCA for preprocessing the baseline models, we tune the number of principal components (PCs) used to represent the neural images, treating it as a hyperparameter. We select the number of PCs from the set of values $\{25, 50, 100, 200, 400\}$. After training baseline models with PCA preprocessing and obtaining predictions of PCs in the low-dimensional space, we project the predictions back to the neural image space to compare the performance in neural prediction with SBIND.

### A.3.2. LocaNMF

Localized semi-nonnegative matrix factorization (LocaNMF) (Saxena et al., 2020) is a dimensionality reduction method that decomposes widefield imaging data into localized spatial components and corresponding temporal components. The temporal component is used as a low-dimensional representation of widefield calcium imaging data. LocaNMF leverages the Allen brain atlas (Wang et al., 2020) to initialize spatial components and encourages localization by limiting their spread, while still allowing contributions from neighboring regions to capture relevant variance. The result is a more interpretable decomposition, where each temporal component primarily corresponds to a specific brain region

When using LocaNMF for preprocessing, we tune its hyperparameters. The number of components for Singular Value Decomposition is varied within the set of values $\{100, 200, 400, 1000\}$. The "minrank" hyperparameter is selected from the set $\{1, 2, 5\}$. Other hyperparameters are set to their default values.

### A.3.3. CEBRA

CEBRA (Schneider et al., 2023) is a non-dynamic model that uses convolutional neural network encoders in its architecture to process neighboring time points of the data within a small, fixed window length. It uses a contrastive loss to extract latent embeddings informed by either simultaneous behavior labels or time information. CEBRA-Behavior uses an objective that aligns neural activity in the embedding space such that time points with similar behavior have similar embeddings. CEBRA-Time uses time information to extract the embeddings. After learning the embeddings, it fits decoders to map the embeddings from each time point to the observation space (i.e., behavior or neural).

We performed hyperparameter tuning for CEBRA. We used the default "KNN-Decoder" for categorical behavior prediction in WFCI 2 and fUSI datasets. For neural prediction and continuous behavior, we tested the default "KNN-Decoder", "L1 Linear Regressor", and Ridge regressor decoder, with the latter achieving superior performance. We report neural reconstruction (zero-step-ahead) across folds and all the pixels within the brain areas for CEBRA. For continuous behavioral data, we report same-step decoding performance, and for categorical behavioral data, we report accuracy, auc, or F1-score as appropriate. The embedding dimension was explored from 1 to 256 in powers of 2. For widefield datasets, the "time-offset" was selected from $\{5, 10, 20\}$, while for the functional ultrasound imaging dataset, it was chosen from $\{3, 5, 10\}$. The "temperature" hyperparameter was varied within the set $\{0.01, 0.1, 1, 10\}$. The best-performing model across folds is reported for all experiments.

### A.3.4. DPAD

DPAD (Dissociative Prioritized Analysis of Dynamics) (Sani et al., 2024) is a nonlinear dynamical model that focuses on learning behaviorally relevant neural dynamics and dissociating them from other dynamics in neural activity. It achieves this by fitting two dynamical models, formulated as a two-section RNN, one for behaviorally relevant neural dynamics and another for the remaining neural dynamics. DPAD also replaces linear mappings in the dynamical models with MLPs to flexibly learn the source of nonlinearity in the data. However, it is not specifically designed for image data and thus does not explicitly account for the spatial structure in the image-distributed neural data.

To compare with DPAD, we first identified the source of nonlinearity by using MLPs for each of the parameters of the model. We identified behavior readout parameter, $C_z$, as source of nonlinearity and used an MLP with 1 or 2 hidden layers for this parameter. We tuned DPAD hyperparameters by varying the latent state dimension from 1 to 256 in powers of 2. For neural prediction and continuous behavioral data, we report one-step-ahead prediction for comparison. For categorical data, we report accuracy, AUC, or F1-score as appropriate.

### A.3.5. STNDT

STNDT (Le & Shlizerman, 2022) utilizes a Transformer architecture for spatiotemporal modeling of neural population spiking activity. Originally designed for spiking data, we adapted STNDT to accept preprocessed LocaNMF features as input, as its direct application to raw images is computationally prohibitive due to the quadratic complexity of its spatial self-attention mechanism over a large number of pixels. For these LocaNMF features, we placed a Gaussian prior on the components and employed an MSE loss instead of the model's original Poisson likelihood. STNDT was trained using its original objectives, including masked reconstruction and a contrastive loss. For behavior decoding, we followed the approach discussed on STNDT's OpenReview forum, which is to use ridge regression to decode behavior from the learned latent states.

For hyperparameter tuning when using STNDT with LocaNMF features, we used the default hyperparameter choices such as number of transformers, masking ratio, etc., and varied the number of LocaNMF components provided as input (i.e., embedding dimension for STNDT), exploring values in the set $\{55, 123, 270\}$. Table A.5 reports performance of the adapted STNDT model which achieves the best behavior decoding.

### A.3.6. TNDM

TNDM (Hurwitz et al., 2021) is a sequential autoencoder-based model designed to learn two distinct sets of latent factors from spiking data, with dimensionalities $n_1$ and $n_2$, corresponding to behaviorally relevant and behaviorally irrelevant dynamics, respectively. It achieves this by optimizing a combined neural-behavioral reconstruction loss. Given TNDM's original design for Poisson-distributed spiking data, we adapted it for our widefield imaging datasets. This involved using preprocessed LocaNMF features as input, assuming a Gaussian distribution for these input features, and changing TNDM's neural reconstruction loss to MSE.

Hyperparameter tuning for TNDM involved sweeping the dimensionalities for the behaviorally relevant latent factors, $n_1$, selected from $\{8, 16, 32, 64\}$, and the behaviorally irrelevant latent factors, $n_2$, selected from $\{0, 8, 16, 32, 64\}$. Table A.5 reports performance of the adapted TNDM model which achieves the best behavior decoding.

## A.4. Datasets Details

### A.4.1. WIDEFIELD CALCIUM IMAGING (WFCI) DATASETS

The WFCI datasets were collected from head-fixed mice performing a decision-making task, where they reported the spatial position of auditory or visual stimuli by licking the corresponding spout (Churchland et al., 2019). Neural activity across the dorsal cortex was optically recorded at 30 and 15 Hz for WFCI 1 and WFCI 2 datasets, respectively. We preprocessed the neural images to remove hemodynamic artifacts using a linear regression method (Musall et al., 2019; Valley et al., 2020). The raw neural images, with dimensions 540x640 pixels, were cropped to include only the brain regions and downsampled to 128x128 pixels. A pixel-wise temporal causal filter (0.1 Hz, 2nd order Butterworth high-pass) was applied to remove drift in the time series.

**WFCI 1:** This dataset consists of 248 trials with variable lengths ($6.59 \pm 0.50$ seconds). Concurrently with neural recordings, behavior videos were recorded from two viewpoints (face and bottom; see Figure 2a). Following a similar procedure to (Musall et al., 2019), 14 dimensions of continuous behavior were extracted from seven regions of interest (eye, nose, whisker, paw, chest, body, and mouth) in the videos (see Figure 2c). For each region, the first principal component of both the original video frames and the motion video frames (computed as the absolute temporal derivative of frames) was extracted and used for behavioral prediction (see Figure A.5).

**WFCI 2:** This dataset comprises 412 trials with variable lengths ($6.30 \pm 0.37$ seconds) with the same trial structure and neural recordings as WFCI 1. However, instead of behavior videos, four binary sensors detected contact with the animal's forepaws (handles) and tongue (spouts), providing categorical behavioral data for decoding (see Figure 2d), where 1's in any of the binary traces represents the time samples where the mouse was touching the corresponding sensor.

### A.4.2. FUNCTIONAL ULTRASOUND IMAGING (fUSI) DATASET

The fUSI dataset consists of recordings from a non-human primate performing a memory-guided saccade or reach task to either 2 or 8 peripheral targets (Griggs et al., 2023). Trials began with a $5 \pm 1$ second fixation period, followed by a 400 ms presentation of a peripheral cue. After the cue disappeared, there was a $5 \pm 1$ second memory period before the monkey executed a saccade or reach to the remembered target location. Successful trials were followed by a $1.5 \pm 0.5$ second hold period and then a reward. Each trial was followed by an inter-trial period before the next trial started.

**Preprocessing:** We applied a causal temporal voxel-wise filter (0.02 Hz high-pass Butterworth, 2nd order) to remove drift during the sessions. Similar to (Griggs et al., 2024), we z-scored the data voxel-wise over a rolling 60-frame buffer. Next, a pillbox spatial filter with a radius of 2 pixels was applied to each frame. The images were originally $128 \times 132$ pixels and cropped to $128 \times 128$ pixels.

**Decoding:** In this dataset, behavior consisted of the target the monkey reached or fixated on for each trial. Thus, we considered behavior as available only during the 1.5-second period (equivalent to 3 samples) before reward period when the monkey was fixating on the target. This gave us a categorical and intermittently recorded behavior time-series for modeling. We used these 3 samples as the only samples in the trials of length 30 where we have intermittent behavior available. To fit all variants of SBIND, we masked out other time points in the trial and optimized the parameters only for those 3 specific samples, effectively implementing intermittent behavior decoding during training. In the 2-directional tasks, we used binary target classification. In 8-directional tasks, similar to (Griggs et al., 2024), we used a multi-decoder approach in the decoder mapping to predict the vertical and horizontal directions. Thus, the decoder $D^{(1)}$ has 6 softmaxed output dimensions: 3 for probabilities of left-right-center summing to 1, and 3 for probabilities of up-center-down summing to 1. For training PCA+LDA, we used either the 3 samples before the reward period, as in (Griggs et al., 2024), or all samples of the trial to decode directions. For training CEBRA, we used either the 3 samples before the reward period or all the samples (default choice for target classification task) of the trials for learning the embeddings, with the latter proving more effective for the target classification task. To fit DPAD, we used the 3 samples before reward period to fit the first RNN. During evaluation, we used the latent embedding at the last time-step in the trial to predict the direction of movement.

## A.5. Supplementary Experiments

*Table A.1.* One-step-ahead behavior decoding and neural prediction performances for various ablations of SBIND across 5 folds for WFCI 1 dataset in terms of $R^2$. As indicated by the arrows, higher is better for $R^2$. For neural prediction, $R^2$ (Mean $\pm$ SEM) is reported across 5 folds and all pixels within the brain areas. For behavior decoding, $R^2$ (Mean $\pm$ SEM) is reported across 5 folds and 14 dimensions of behavior.

| MODEL | PREPROCESSING | BEH. $R^2 \uparrow$ | NEUR. $R^2 \uparrow$ |
|---|---|---|---|
| MLP-SBIND | FLATTEN | $0.3620 \pm 0.0194$ | $0.8209 \pm 0.0033$ |
| MLP-SBIND | LOCANMF | $0.4025 \pm 0.0152$ | $0.8702 \pm 0.0015$ |
| MLP-SBIND | PCA | $0.3934 \pm 0.0145$ | $\mathbf{0.8926 \pm 0.0016}$ |
| SBIND-UNSUP | - | $0.4589 \pm 0.0108$ | $0.8724 \pm 0.0039$ |
| SBIND NOATT | - | $0.4612 \pm 0.0107$ | $0.8652 \pm 0.0032$ |
| SBIND W/O $f_A$ | - | $0.2080 \pm 0.0453$ | $0.8543 \pm 0.0037$ |
| SBIND MSE $L_Y$ | - | $\mathbf{0.5030 \pm 0.0179}$ | $0.8217 \pm 0.0133$ |
| SBIND | - | $\mathbf{0.5059 \pm 0.0166}$ | $\mathbf{0.8724 \pm 0.0069}$ |

*Table A.2.* One-step-ahead behavior decoding and neural prediction performances for various ablations of SBIND across 5 folds for WFCI 2 dataset in terms of $R^2$ and AUC. As indicated by the arrows, higher is better for $R^2$ and AUC. For neural prediction, $R^2$ (Mean $\pm$ SEM) is reported across 5 folds and all pixels in the brain areas. For behavior decoding, AUC (Mean $\pm$ SEM) is reported across 5 folds and 4 classification tasks for left handle, right handle, left spout, and right spout.

| MODEL | PREPROCESSING | BEH. AUC$\uparrow$ | NEUR. $R^2 \uparrow$ |
|---|---|---|---|
| MLP-SBIND | FLATTEN | $0.8706 \pm 0.0038$ | $0.7503 \pm 0.0076$ |
| MLP-SBIND | LOCANMF | $0.8823 \pm 0.0037$ | $0.6402 \pm 0.0041$ |
| MLP-SBIND | PCA | $0.8120 \pm 0.0325$ | $0.6703 \pm 0.0239$ |
| SBIND-UNSUP | - | $0.9182 \pm 0.0039$ | $\mathbf{0.7970 \pm 0.0133}$ |
| SBIND NOATT | - | $0.9071 \pm 0.0060$ | $0.7451 \pm 0.0043$ |
| SBIND W/O $f_A$ | - | $0.8934 \pm 0.0045$ | $0.7339 \pm 0.0022$ |
| SBIND MSE $L_Y$ | - | $\mathbf{0.9299 \pm 0.0029}$ | $0.7418 \pm 0.0049$ |
| SBIND | - | $\mathbf{0.9282 \pm 0.0020}$ | $\mathbf{0.7749 \pm 0.0074}$ |

*Table A.3.* Behavior decoding and neural prediction $R^2$ (Mean $\pm$ SEM) across folds for WFCI 1 dataset. As indicated by the arrows, higher is better for $R^2$. For neural prediction, $R^2$ (Mean $\pm$ SEM) is reported across 5 folds and all pixels in the brain areas. For behavior decoding, $R^2$ (Mean $\pm$ SEM) is reported across 5 folds and 14 dimensions of behavior.

| MODEL | PREPROCESSING | BEH. $R^2 \uparrow$ | NEUR. $R^2 \uparrow$ |
|---|---|---|---|
| DPAD | FLATTEN | $0.3826 \pm 0.0189$ | $0.8434 \pm 0.0022$ |
| DPAD | LOCANMF | $0.4128 \pm 0.0133$ | $0.8697 \pm 0.0011$ |
| DPAD | PCA | $0.3839 \pm 0.0157$ | $\mathbf{0.8902 \pm 0.0008}$ |
| CEBRA | FLATTEN | $0.4001 \pm 0.0132$ | $0.5957 \pm 0.0216$ |
| CEBRA | LOCANMF | $0.3745 \pm 0.0081$ | $0.4453 \pm 0.0099$ |
| CEBRA | PCA | $0.3686 \pm 0.0127$ | $0.4638 \pm 0.0079$ |
| SBIND | - | $\mathbf{0.5059 \pm 0.0166}$ | $\mathbf{0.8724 \pm 0.0069}$ |

*Table A.4.* Behavior decoding AUC and neural prediction $R^2$ (Mean $\pm$ SEM) across folds for WFCI 2 dataset. As indicated by the arrows, higher is better for $R^2$ and AUC. For neural prediction, $R^2$ (Mean $\pm$ SEM) is reported across 5 folds and all pixels in the brain areas. For behavior decoding, AUC (Mean $\pm$ SEM) is reported across 5 folds and 4 classification tasks for left handle, right handle, left spout, and right spout. For CEBRA, a "KNNDecoder" is used for decoding which does not directly report AUC.

| MODEL | PREPROCESSING | BEH. AUC$\uparrow$ | NEUR. $R^2\uparrow$ |
|---|---|---|---|
| DPAD | FLATTEN | $0.8782 \pm 0.0059$ | $0.7440 \pm 0.0049$ |
| DPAD | LOCANMF | $0.8888 \pm 0.0057$ | $0.6374 \pm 0.0033$ |
| DPAD | PCA | $0.8039 \pm 0.0090$ | $0.7182 \pm 0.0038$ |
| CEBRA | FLATTEN | - | $0.7228 \pm 0.0052$ |
| CEBRA | LOCANMF | - | $0.4971 \pm 0.0045$ |
| CEBRA | PCA | - | $0.4913 \pm 0.0131$ |
| SBIND | - | $\mathbf{0.9282 \pm 0.0020}$ | $\mathbf{0.7749 \pm 0.0074}$ |

*Table A.5.* Comparison of baselines including adapted STNDT and TNDM on WFCI1 dataset. Behavior decoding and neural prediction MSE and $R^2$ (Mean $\pm$ SEM) across folds.

| MODEL | PREPROCESSING | NEUR. MSE $\downarrow$ | NEUR. $R^2 \uparrow$ | BEH. MSE $\downarrow$ | BEH. $R^2 \uparrow$ |
|---|---|---|---|---|---|
| DPAD | LOCANMF | $0.0543 \pm 0.0009$ | $0.8697 \pm 0.0011$ | $0.5877 \pm 0.0226$ | $0.4128 \pm 0.0133$ |
| CEBRA | LOCANMF | $0.4976 \pm 0.0241$ | $0.4453 \pm 0.0099$ | $0.6250 \pm 0.0194$ | $0.3745 \pm 0.0081$ |
| STNDT | LOCANMF | $0.0685 \pm 0.0090$ | $0.8376 \pm 0.0088$ | $0.6033 \pm 0.0240$ | $0.3951 \pm 0.0156$ |
| TNDM | LOCANMF | $0.7912 \pm 0.0290$ | $0.5022 \pm 0.0081$ | $0.7749 \pm 0.0240$ | $0.2233 \pm 0.0109$ |
| SBIND-UNSUP | - | $\mathbf{0.0403 \pm 0.0020}$ | $\mathbf{0.8724 \pm 0.0039}$ | $0.5413 \pm 0.0185$ | $0.4589 \pm 0.0108$ |
| SBIND | - | $\mathbf{0.0414 \pm 0.0029}$ | $\mathbf{0.8724 \pm 0.0069}$ | $\mathbf{0.4955 \pm 0.0254}$ | $\mathbf{0.5059 \pm 0.0166}$ |

*Table A.6.* Performance comparison of SBIND on WFCI1 datasets using two recurrent update formulations for integrating current neural image information ($\mathbf{Y}_k$). The table contrasts the concatenation approach as in Eq. A.12, where the encoded input $\boldsymbol{K}(\mathbf{Y}_k)$ is concatenated with the latent state $\mathbf{X}_k$ before the recurrent function, against the summation approach as in Eq. A.1. Results demonstrate that the concatenation method Eq. A.12 yields improved neural prediction MSE, neural $R^2$, behavioral MSE, and behavioral $R^2$. for WFCI1 dataset.

| Model Formulation | Neural MSE | Neural $R^2$ | Beh MSE | Beh $R^2$ |
|---|---|---|---|---|
| SBIND w. Recurrent Eq. A.12 | $0.0414 \pm 0.0029$ | $0.8724 \pm 0.0069$ | $0.4955 \pm 0.0254$ | $0.5059 \pm 0.0166$ |
| SBIND w. Recurrent Eq. A.1 | $0.0664 \pm 0.0013$ | $0.8545 \pm 0.0027$ | $0.5306 \pm 0.0198$ | $0.4680 \pm 0.0182$ |

*Table A.7.* Comparison of various ablations in behavior decoding accuracy (quantified as proportion of trials whose target was correctly decoded) and AUC across 10 folds and all sessions of fUSI Data. For 8-directional sessions a multi-decoder approach is used with two decoders to predict vertical and horizontal directions (left-right-stationary). Multi-class AUC averaged over two vertical and horizontal directions, 4 sessions and 10 folds are reported. As indicated by the arrows, higher is better for accuracy and AUC.

| MODEL | PREPROCESSING | 2-DIRECTIONAL SESSIONS | | 8-DIRECTIONAL SESSIONS | |
|---|---|---|---|---|---|
| | | BEH. ACCURACY$\uparrow$ | BEH. AUC$\uparrow$ | BEH. ACCURACY$\uparrow$ | BEH. AUC$\uparrow$ |
| MLP-SBIND | FLATTEN | $0.5984 \pm 0.0162$ | $0.6285 \pm 0.0216$ | $0.2175 \pm 0.0169$ | $0.5833 \pm 0.0196$ |
| MLP-SBIND | PCA | $0.6565 \pm 0.0178$ | $0.7296 \pm 0.0214$ | $0.2966 \pm 0.0183$ | $0.6823 \pm 0.0170$ |
| SBIND-UNSUP | - | $0.7030 \pm 0.0180$ | $0.7859 \pm 0.0194$ | $0.3411 \pm 0.0187$ | $0.7304 \pm 0.0176$ |
| SBIND NOATT | - | $0.6889 \pm 0.0184$ | $0.7480 \pm 0.0237$ | $0.2893 \pm 0.0179$ | $0.6975 \pm 0.0145$ |
| SBIND | - | $\mathbf{0.7300 \pm 0.0191}$ | $\mathbf{0.8067 \pm 0.0180}$ | $\mathbf{0.3521 \pm 0.0201}$ | $\mathbf{0.7393 \pm 0.0169}$ |

*Table A.8.* Comparison of various ablations of SBIND in one-step-ahead neural prediction performance (MSE and $R^2$) across 10 folds and all sessions of fUSI dataset. As indicated by the arrows, lower is better for MSE and higher is better for $R^2$.

| MODEL | PREPROCESSING | 2-DIRECTIONAL SESSIONS | | 8-DIRECTIONAL SESSIONS | |
|---|---|---|---|---|---|
| | | NEUR. MSE↓ | NEUR. $R^2$↑ | NEUR. MSE↓ | NEUR. $R^2$↑ |
| MLP-SBIND | FLATTEN | $0.7885 \pm 0.0067$ | $0.2641 \pm 0.0060$ | $0.8743 \pm 0.0065$ | $0.2274 \pm 0.0071$ |
| MLP-SBIND | PCA | $0.6322 \pm 0.0054$ | $0.3985 \pm 0.0043$ | $0.7058 \pm 0.0030$ | $0.3586 \pm 0.0038$ |
| SBIND-UNSUP | - | $\mathbf{0.4453 \pm 0.0127}$ | $\mathbf{0.6029 \pm 0.0112}$ | $\mathbf{0.3827 \pm 0.0086}$ | $\mathbf{0.6683 \pm 0.0073}$ |
| SBIND NOATT | - | $0.5036 \pm 0.0155$ | $0.5545 \pm 0.0133$ | $0.4230 \pm 0.0109$ | $0.6344 \pm 0.0090$ |
| SBIND MSE $L_{\mathbf{Y}}$ | - | $0.4900 \pm 0.0108$ | $0.5555 \pm 0.0094$ | $0.4225 \pm 0.0076$ | $0.6287 \pm 0.0064$ |
| SBIND | - | $\mathbf{0.4725 \pm 0.0165}$ | $\mathbf{0.5736 \pm 0.0144}$ | $\mathbf{0.3919 \pm 0.0107}$ | $\mathbf{0.6558 \pm 0.0094}$ |

*Table A.9.* Comparison of baselines in behavior decoding AUC and neural prediction $R^2$ across 10 folds and all sessions (Mean $\pm$ SEM). For DPAD and SBIND, in 2-directional sessions the AUC for binary classification is reported over 10 folds and 9 sessions of fUSI Data. In 8-directional sessions, a multi-decoder approach is used with two decoders to predict vertical and horizontal directions (left-right-stationary). Multi-class AUC averaged over two vertical and horizontal directions, 4 sessions, and 10 folds are reported. For CEBRA, a "KNNDecoder" is used for decoding which does not directly report AUC. As indicated by the arrows, higher is better for AUC and $R^2$.

| MODEL | PREPROCESSING | 2-DIRECTIONAL SESSIONS | | 8-DIRECTIONAL SESSIONS | |
|---|---|---|---|---|---|
| | | BEH. AUC↑ | NEUR. $R^2$↑ | BEH. AUC↑ | NEUR. $R^2$↑ |
| LDA | PCA | $0.7130 \pm 0.0242$ | - | $0.6987 \pm 0.0164$ | - |
| DPAD | FLATTEN | $0.5940 \pm 0.0221$ | $0.2355 \pm 0.0066$ | $0.6060 \pm 0.0157$ | $0.2011 \pm 0.0079$ |
| DPAD | PCA | $0.7399 \pm 0.0208$ | $0.3938 \pm 0.0045$ | $0.6752 \pm 0.0179$ | $0.3554 \pm 0.0040$ |
| CEBRA | FLATTEN | - | $-0.3839 \pm 0.0128$ | - | $-0.3518 \pm 0.0085$ |
| CEBRA | PCA | - | $-0.2731 \pm 0.0069$ | - | $-0.3076 \pm 0.0103$ |
| SBIND | - | $\mathbf{0.8067 \pm 0.0180}$ | $\mathbf{0.5736 \pm 0.0144}$ | $\mathbf{0.7393 \pm 0.0169}$ | $\mathbf{0.6558 \pm 0.0094}$ |

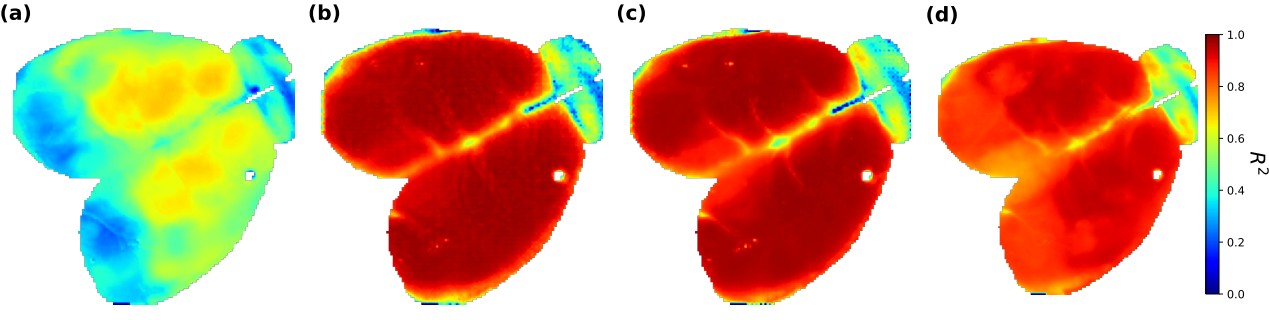

*Figure A.2.* **Pixel-wise neural prediction $R^2$ values across brain regions.** Neural prediction maps for the WFCI 1 dataset are depicted for **(a)** CEBRA, **(b)** SBIND, **(c)** SBIND NoAtt, and **(d)** DPAD. SBIND produces more detailed neural predictions compared to its variant without the self-attention mechanism. CEBRA poorly predicts neural activity because it uses latent embeddings guided by behavior and lacks extra embedding dimensions for residual neural activity unrelated to behavior. This is consistent with the observation that widefield calcium imaging datasets often contain significant neural activity unrelated to behavior (Musall et al., 2019). Higher is better for $R^2$.

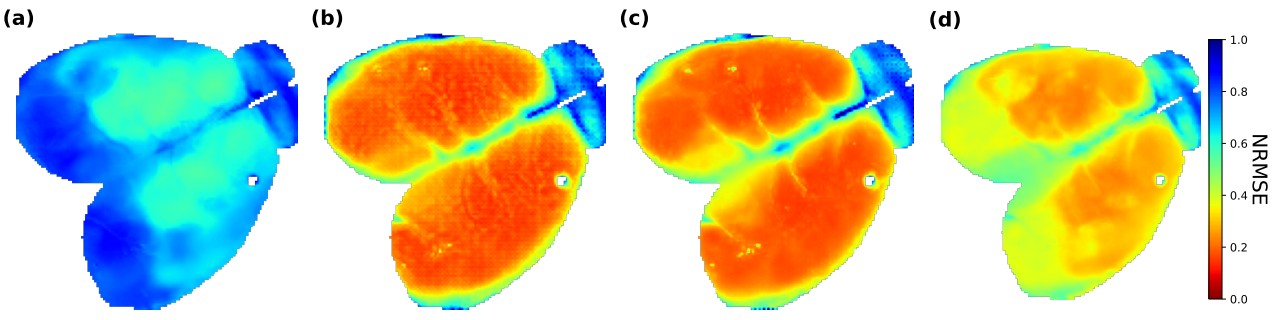

*Figure A.3.* **Pixel-wise neural prediction NRMSE values across brain regions.** Neural prediction maps for the WFCI 1 dataset are depicted for **(a)** CEBRA, **(b)** SBIND, **(c)** SBIND NoAtt, and **(d)** DPAD. Lower is better for NRMSE.

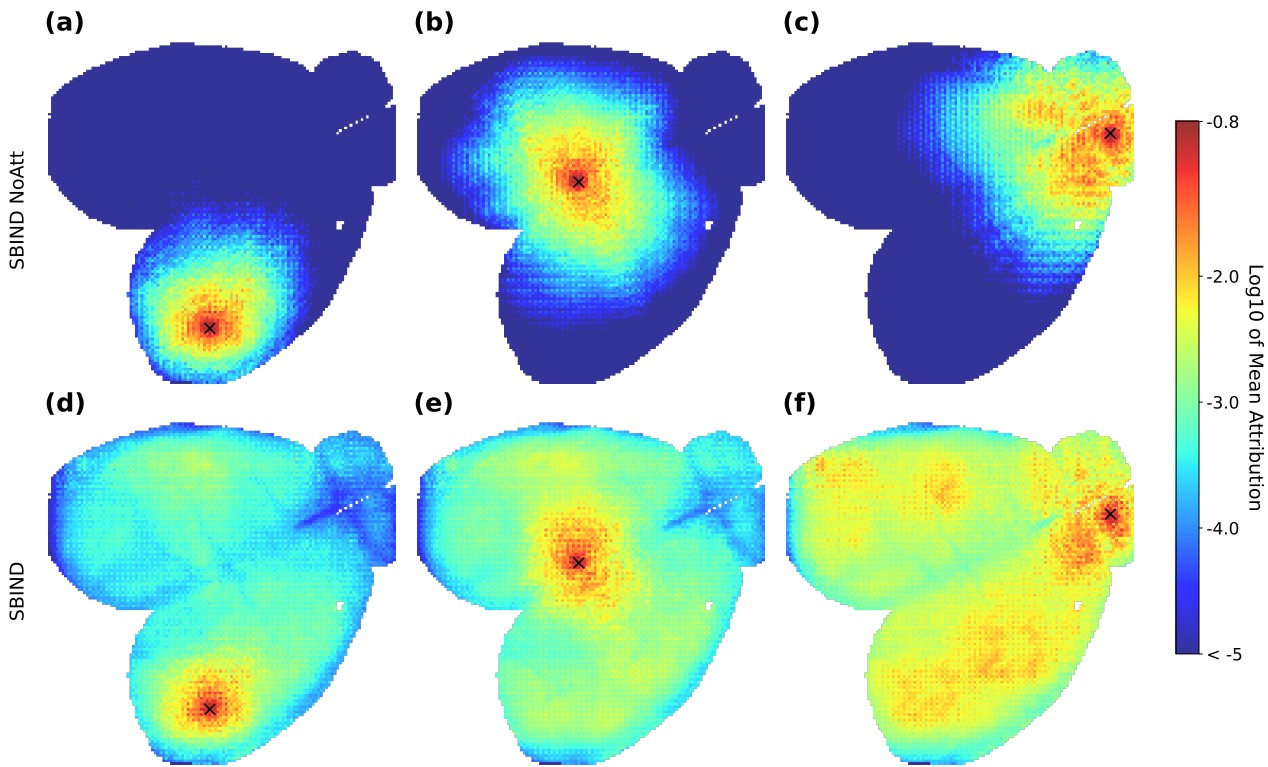

*Figure A.4.* **Mean contribution of all brain regions to predicting the activity of three different pixels marked by × in the brain map.** The plots display the mean attribution of whole-brain activity to the neural prediction of specific points, derived using the Captum framework. We calculated the attribution of each input image across all time points to the neural prediction of the specified points in different plots. These attribution maps were then averaged across time for different neural images to find the mean attribution. This analysis was performed on the test data from the WFCI 1 dataset after training both models. **(a-c)** SBIND NoAtt. **(d-f)** SBIND.

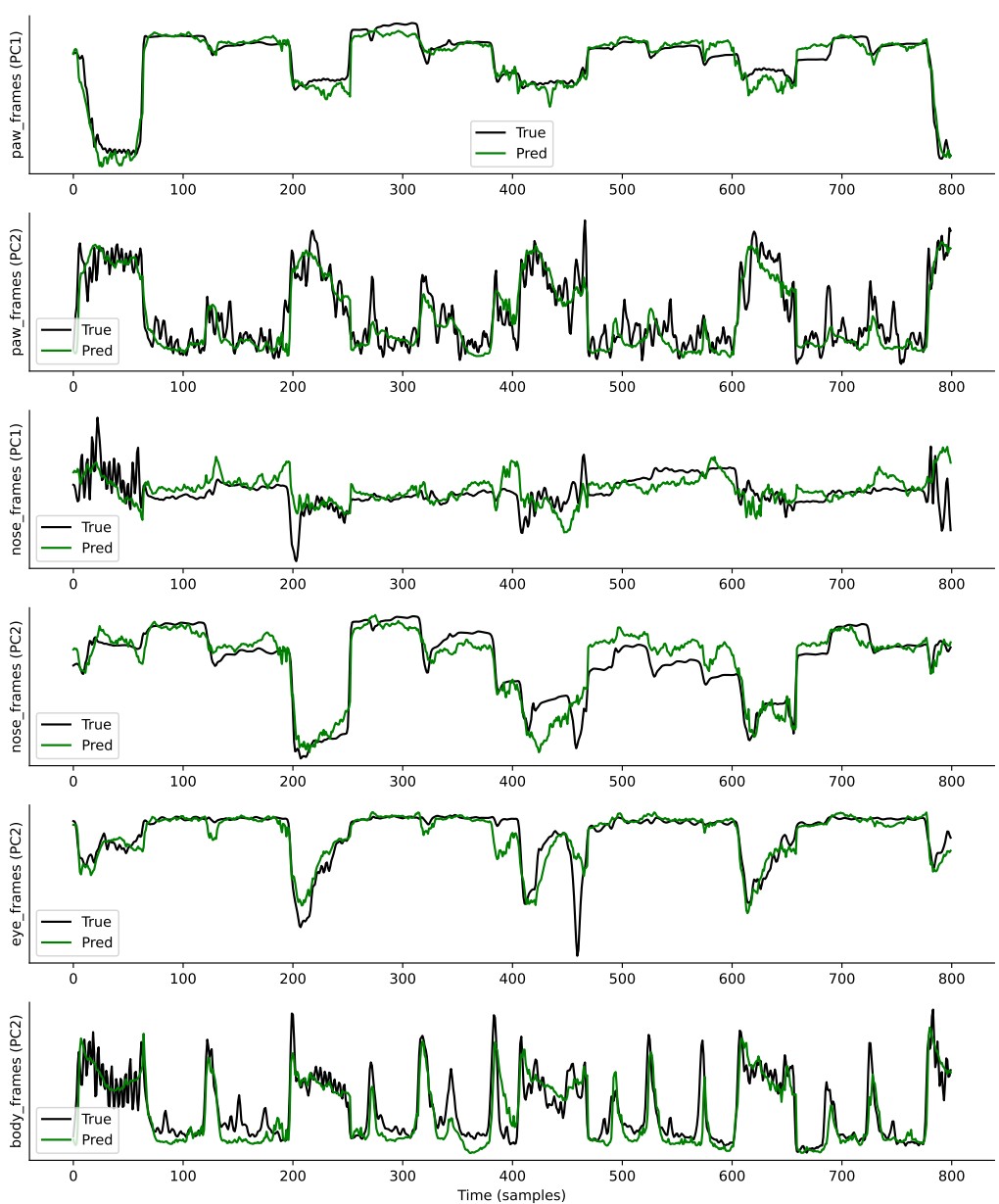

*Figure A.5.* **SBIND example behavior decoding.** Predictions over 6 dimensions of continuous behavior extracted from behavior videos for WFCI 1 dataset. (See Appendix A.4 and Figure 2 for details of behavior extraction.)

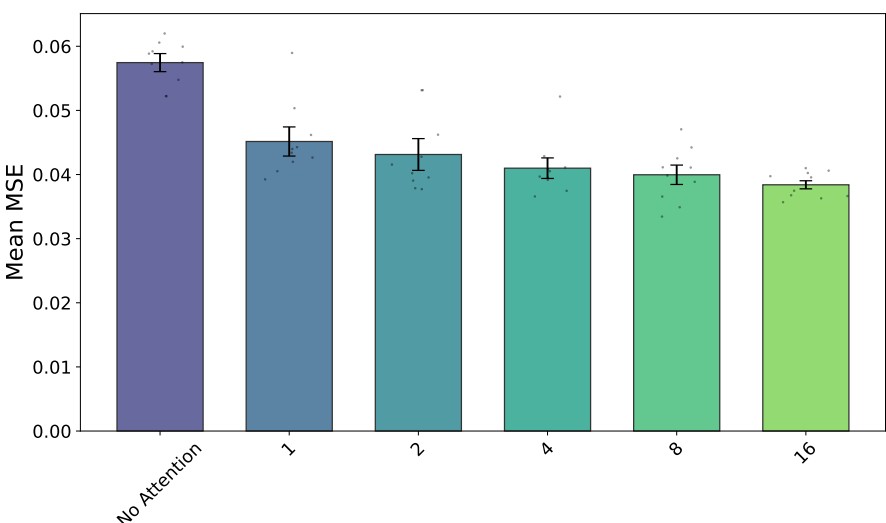

*Figure A.6.* **Larger Self-Attention Patch Sizes Lead to Better Neural Prediction.** Neural prediction MSE across different patch sizes for the WFCI 1 dataset across 5 folds and 2 runs.

