# OpenReview forum: "Dynamical Modeling of Behaviorally Relevant Spatiotemporal Patterns in Neural Imaging Data"
_ICML.cc/2025/Conference — ICML 2025 poster_

### Official Review · Reviewer_gpPL · 2025-03-12

**Overall Recommendation:** 3

**Summary:**

This work proposed a novel learning framework, SBIND, to model spatiotemporal neural imaging. SBIND is able to model the spatiotemporal neural dynamics of neural activity, at the same time SBIND can disentangle the behavioral relevant neural dynamics. The authors experimented with both calcium and ultrasound imaging datasets. Further model comparison with baselines demonstrates the proposed method is able to decode the behavior data and recover the neural activity from the latent space with reasonable accuracy. Ablation studies were also conducted to prove the effectiveness of the proposed method.

**Claims And Evidence:**

The claims are reasonable.

**Essential References Not Discussed:**

The submission includes sufficient references as far as I know.

**Experimental Designs Or Analyses:**

The experimental designs are reasonable. How the hyperparameter search is performed?

**Methods And Evaluation Criteria:**

The motivation behind taking ConvRNN1 outputs as inputs to ConvRNN2 is unclear.

**Other Comments Or Suggestions:**

On lines 88-88, the name, CovRNN, is confusing: "utilized convolutional recurrent neural networks (CovRNN)," while the model temporal relationship is learned with attention instead of RNN.

In section 2.2, when mentioning ConvRNN1 it is better to use ConvRNN1 or ConvRNN2 to avoid confusion.

**Other Strengths And Weaknesses:**

The authors proposed a novel method to learn behavioral-relevant neural representations and neural dynamics at the same time. The method can be applied to a general neural imaging dataset. The performance improvement is significant compared to previous methods. Besides, the model architecture designed simultaneously learns the behavioral relevant dynamics and the general dynamics which can help with understanding of neural functions.


It will also be good to see how the proposed method can be applied to electrode-based neural recordings and compare the results with the baseline methods. I believe technically, the approach can be applied to various data formats.

**Questions For Authors:**

In the method section, what is $n_y$, $n_x$, $n_z$ corresponding to?

Is that necessary to add ConvRNN1 outputs to ConvRNN2, I think there will be redundant information.

How large the current dataset is? It seems the number of available trials is limited. Did the authors see any overfitting issues?

What do the authors think are the main components that contribute most to performance improvement?

For the $f_A$ model authors mentioned the function is for learning long temporal relationships, how authors concatenate image patches with the temporal steps. How the number of patches and the length of temporal window will affect the results.

**Relation To Broader Scientific Literature:**

Previous literature learns the behavioral relevant neural dynamics and behavioral-irrelevant neural dynamics in two stages, while SBIND can learn them simultaneously in one stage. Also the

**Theoretical Claims:**

No theoretical claims were made in the submission.

---

> ### Author Rebuttal · Authors · 2025-04-01
>
> ### [1]: motivation for ConvRNNs
> We thank the reviewer for this insightful question. There are two reasons we need to pass the ConvRNN1 states to ConvRNN2. First, ConvRNN2 aims to capture **residual neural dynamics** not explained by behaviorally relevant states, $X_k^{(1)}$. To ensure $X_k^{(2)}$ learns dynamics not included in $X_k^{(1)}$, we pass $X_k^{(1)}$ to ConvRNN2. Thus, this connection functions similarly to a **residual connection** which avoids re-learning the dynamics of ConvRNN1. Second, Sani et al. 2021 have proven that for linear dynamical systems, to get a general disentangled linear state-space model, one needs a link from $X_{k+1}^{(1)}$ to inform the recursion of $X_k^{(2)}$ (i.e. how $X_{k+1}^{(2)}$ is derived from $X_k^{(2)}$). Although SBIND uses nonlinear ConvRNNs, this linear theory provides another motivation for passing the behaviorally relevant states from ConvRNN1 to ConvRNN2.
>
> ### [2]: Hyperparameter (HP) Search
> Primarily, we swept the latent dimensions $n_x$ and patch size for all datasets, as these were found to be the most critical HPs for performance. Other major HPs (e.g., learning rate, sequence length, convolutional layers were fixed based on the best validation performance on the WFCI datasets. For the fUSI dataset these HPs were picked based on best behavior prediction in the validation set using one representative session and then fixed at these values across all sessions.
>
> ### [3]: Applying to Electrophysiological (EP) Data
> We thank the reviewer for the insightful suggestion to assess SBIND's robustness on EP data. While SBIND is primarily designed for the unique challenges of neural imaging, we agree this is a valuable comparison. We applied SBIND to an **EP dataset** (O'Doherty et al., 2020) containing smoothed spike counts from 42 channels and 4D behavioral kinematics (horizontal and vertical velocity and position) from two sessions of NHP reaching movements. We did not have access to the spatial coordinates of the electrodes, so we could not form an image based on true coordinates. Instead, for comparison purposes, we formed an **arbitrary** 6x7 2D pseudo-image from the 42 channels.
>
> Despite this the pseudo-image input lacking true spatial structure, SBIND performed comparably to DPAD (**new Table B.4**), suggesting the robustness in SBIND's dynamical modeling and disentanglement approach. Finally, as the reviewer noted, we agree SBIND holds potential for other imaging modalities or modalities that naturally possess a grid structure, such as EEG recordings.
>
> **Table B.4: Comparison with DPAD on EP Recordings**
> |Model|Neur. R2|Beh. R2|
> |-|-|-|
> |DPAD|0.8321±0.0089|0.5053±0.0201|
> |SBIND|0.8247±0.0079|0.4972±0.0212|
>
> ---
> ### [4]: Comments & Suggestions
> We thank the reviewer for the helpful suggestions to improve clarity:
>
> * **ConvRNN naming**: SBIND employs an **attention-augmented ConvRNN** architecture. While the self-attention (SA) module captures *spatial* dependencies within each time step, the architecture remains fundamentally *recurrent* because the state equation ensures that $X_{k+1}$ depends on $X_{k}$. Therefore, we used "ConvRNN" to describe the overall model class, with the understanding that it incorporates SA. We will clarify this in the manuscript and ensure "ConvRNN1" and "ConvRNN2" are used consistently.
>
> * **$n_y, n_x, n_z$ definitions**: Please refer to Reviewer z1x2 response [1].
>
> ### [5]: Dataset Size
> WFCI 1 and WFCI 2 contain 49k, and 39k frames. While the fUSI dataset has ~5k frames per session, the availability of 13 sessions provides enough data overall for evaluation. To mitigate overfitting, we employed standard techniques including L2 weight decay, dropouts, and early stopping. Example loss curves are provided via [this anonymous link](https://anonymous.4open.science/r/sbind-88FF/loss.png), showing that overfitting was not an issue.
>
> ### [6]: Main Contributing Components
> Ablation studies (Sec 4.2; Tables 1, A.1, A.2; Fig 4) show SBIND's performance benefits stem from its ConvRNN architecture with integrated SA and use of GDL loss for more precise neural dynamical modeling. Also, learning behaviorally relevant dynamics directly from raw images leads to better behavior decoding.
>
> ### [7]: $f_A$ function
> The SA operates purely in the *spatial* domain within a single time step, k. At each time, the latent image $X_{k}$ is divided into spatial patches, and the multi-head SA computes the relationships between these patches *at that specific time* (not across different time steps). The **temporal dependencies** are handled by the outer *recurrent structure* of the ConvRNN, where $X_{k+1}$ is computed based on $X_{k}$. The effect of patch size (which influences the *spatial scope* of attention) was investigated in our ablation study (Figure A.6), showing that larger patch sizes result in better self-prediction. Also, the sequence length HP was chosen to balance sufficient temporal context with the computational limits.
>
> **References:** Line 495

---

### Official Review · Reviewer_z1x2 · 2025-03-12

**Overall Recommendation:** 4

**Summary:**

The authors propose SBIND, which learns behaviorally relevant and irrelevant neural dynamics directly from high-dimensional imaging data without preprocessing. The authors apply SBIND to widefield imaging datasets and functional ultrasound, and find that SBIND outperforms existing methods that involve preprocessing in predicting one-step-ahead behavior and neural activity.

**Claims And Evidence:**

The authors claim that SBIND outperforms existing methods, and this is well demonstrated by Tables 1-3, where the authors compared SBIND not just to other methods across multiple datasets but also to SBIND that has different architectures. This analysis allows identifying which components of the model contribute to good performance. The authors show that using both the local and long-range spatial dependencies in the images for the behavioral and neural activity prediction is important for performance. This is demonstrated in e.g., Figure 4 and Appendix A.4.

**Essential References Not Discussed:**

I couldn’t think of essential references not cited in this paper.

**Experimental Designs Or Analyses:**

The experiments are sound. The authors have done rigorous ablation studies in Tables 1, A.1 and A.2 for calcium images and Tables A.5 and A.6 for ultrasound. The authors also report the hyperparameters used in SBIND in Table A.1.

**Methods And Evaluation Criteria:**

The SBIND method involves training of two different ConvRNNs in two stages. In the first phase, a ConvRNN with a self-attention mechanism is trained so that it is optimized to perform behavior prediction on the next timestep along with neural activity prediction. This allows learning a low-dimensional representation that is behaviorally relevant. In the second phase, another ConvRNN with a similar architecture tries to predict only neural activity, but from both the latent representation it learns and the latent representation from the first ConvRNN. This allows the second ConvRNN to learn a low-dimensional representation that is behaviorally irrelevant. This disentangling of representations is shown to be important in SBIND’s performance (Table 1) and makes sense for this application of decoding behavior from neural activity.

Instead of simply using MSE, the authors use the gradient difference loss (GDL) with L1 and L2 functions, which, the authors note, have been shown to preserve local image structures (Mathieu et al., 2016). This makes sense for this application.

**Other Comments Or Suggestions:**

- Line 87 space typo.
- What do H and W mean in Equation (1)? Please define all symbols used in equations.
- Also, for consistency maybe it could say X_{k+1} given {Y_1, …, Y_k}. For inference, couldn’t you have used all Y_{1:K}?
- The anonymous link to the repository didn’t seem to work for me. Is this an error on my end?

**Other Strengths And Weaknesses:**

A major strength of this work is that it is widely applicable to both calcium imaging and functional ultrasound data. It uses convolutional RNN and attention mechanism to take into account local and global spatial information when doing the prediction of the next image and behavior, which has not been done previously. The clarity of the paper can be improved.

**Questions For Authors:**

1. How is the performance affected if the model is trained not in two stages but all at once? Is it possible to do, and still achieve disentangling of latents? Will this potentially improve training time and also performance?

2. Can the method be used to perform behavioral decoding not just for the next timestep, but also for much later future timesteps? How fast is the inference step? I think mentioning these in the text might be helpful in the context of being useful for BCI.

3. In Equation (2), why is it that X and Y go into different networks and sumed? Could you have made the model so that X and Y both go into f_A?

**Relation To Broader Scientific Literature:**

In contrast to previous papers that involve preprocessing steps (e.g., PCA) and pre-defined ROI (e.g., LocaNMF), SBIND takes the calcium imaging and ultrasound imaging data directly to perform neural activity and behavior prediction. As far as I know, methods that can be applied to fUSI are relatively sparse in neuroscience, and because SBIND is a general model that can be applied to both calcium images and ultrasound, this method may have wide applications.

**Theoretical Claims:**

There were no theoretical claims in this paper.

---

> ### Author Rebuttal · Authors · 2025-04-01
>
> ### [1]: Comments & Suggestions
> Thank you for these helpful suggestions for improving the clarity of our work.
>
> * We will fix the typo on Line 87 and ensure all symbols are clearly defined upon first use. Specifically, $n_y$ is the number of neural images in $Y_k$ at time k, and H and W are the height and width. $n_x$ is the number of dimensions of the latent states, and $n_z$ represents the dimension of the behavior vector, $z_k$. Across all datasets, $n_y=1$, but we include $n_y$ for generality of the formulation.
> * We agree with the suggestion regarding notation consistency and will amend Line 162 to clarify the RNN estimates $X_{k+1}$ given $Y_{1:k}$.
> * **Inference using the full sequence $Y_{1:K}$**: Our model performs inference causally (prediction at time k+1 uses observations up to time k), given its importance for causal neuroscience investigations and real-time BCI applications. Using the full sequence would correspond to non-causal inference (i.e., smoothing), which was not the goal of this work but can be achieved using bidirectional RNNs in future studies.
> * Finally, we have verified that the anonymous link to the code works; the previous issue may have been a temporary glitch.
>
> ---
> ### [2]: Combined Loss
> We adopted the two-stage approach because it allows us to explicitly disentangle the behaviorally relevant latent states $X^{(1)}$. The ConvRNN in Stage 1 is optimized *specifically* for behavior prediction, allowing the first latent state, $X^{(1)}$, to focus on capturing the behaviorally relevant dynamics. Stage 2 then focuses on modeling the *residual* neural dynamics, which are not predictable from $X^{(1)}$. This provides a clear separation/disentanglement of states.
>
> While one-stage training with a mixed neural-behavioral loss is possible, doing so poses a challenge for clear disentanglement. Specifically, in this case, there is no guarantee that any state dimension will just focus on behaviorally relevant or just the residual dynamics. As such, states may actually be mixed up rather than disentangled. Doing so also may create practical challenges. First, the mixed loss optimization will heavily rely on **tuning the relative weights** of the neural and behavior loss terms, which can be difficult and sensitive to hyperparameters. Second, achieving disentanglement with a single combined loss may necessitate incorporating additional specialized loss terms or architectural constraints specifically designed for separation –e.g., KL divergence terms in TNDM (Hurwitz et al., 2021) that we now compare to and show that SBIND outperforms it (see Table B.2); finding the optimal balance for these specialized terms typically requires a potentially sensitive hyperparameter tuning process. In contrast, our two-phase approach avoids this sensitive hyperparameter tuning.
>
> ---
> ### [3]: Multi-step Decoding & Inference Speed
> We thank the reviewer for this important question. We will add these points to the manuscript.
>
> **Multi-step Prediction:** SBIND's recurrent architecture inherently allows for **multi-step** prediction. We can do so with no additional optimization or retraining by performing **recursive forecasting** during inference as follows: instead of feeding the neural observation at the next time step into the neural encoder ($K^{(1)}$), we can feed the model's own one-step-ahead neural image prediction, $\hat{Y}_{k+1}$.
>
> This allows the model to predict the subsequent state $X_{k+2|k}$, behavior $z_{k+2}$, and neural image $Y_{k+2}$ without having the observation at time k+1, and this same process can be iterated further into the future.
>
> **Inference Speed:** A single inference step of the SBIND model takes **17.9 ms** on an NVIDIA RTX 6000 GPU, which is faster than the sampling intervals of WFCI (\~33-67 ms) and fUSI (\~100-500 ms), suggesting potential feasibility for real-time BCI applications (Rabut et al, 2024; Mace et al, 2011).
>
> ---
> ### [4]: Equation 2
> This is another great question. Because $X_k$ and $Y_k$ have **different spatial dimensions** ($H' \times W'$ vs. $H \times W$), we must first *encode* $Y_k$ using $K(\cdot)$ to obtain a representation $K(Y_k)$ with the same spatial dimensions as $X_k$. Afterwards, there are two main options: either **sum** $K(Y_k)$ with the processed state $f_A(X_k)$ (as formulated in Eq. 2, or equivalently, Eq. A.1), or **concatenate** $K(Y_k)$ with the state $X_k$ before applying the recurrent function $f_A(\cdot)$ (as discussed in Appendix A.1.5). For the WFCI datasets reported, we utilized the **concatenation approach** (Eq. A.12), and it yielded better performance as seen in the **new Table B.3**.
>
> **Table B.3: Comparison of SBIND Variants on WFCI1 Dataset.**
> ||Neur. R2|Beh. R2|
> |-|-|-|
> |SBIND w. Recurrent Eq. A.12|0.8724±0.0069|0.5059±0.0166|
> |SBIND w. Recurrent Eq. A.1|0.8545±0.0027|0.4680±0.0182|
>
> **References**:
>
> Rabut et al., Functional ultrasound imaging of human brain activity through transparent. Sci Transl Med. 2024.
>
> Also see Lines 502, 462.

---

> > ### Comment · Reviewer_z1x2 · 2025-04-02
> >
> > I thank the authors for the detailed response to my questions. I will keep my recommendation for acceptance.

---

> > > ### Author Response · Authors · 2025-04-08
> > >
> > > We thank the reviewer for considering our response and confirming their recommendation for acceptance. We are grateful for their insightful comments and questions throughout the review process.

---

### Official Review · Reviewer_9wsc · 2025-03-13

**Overall Recommendation:** 2

**Summary:**

This work propose SBIND, a data-driven deep learning framework to model the spatiotemporal dependencies in the neural image data and behavior data. Existing methods fail to model the dependencies of behaviors and neural dynamics. This work allows modeling the complex local and global spatial temporal patterns, and achieving better performance in predicting dynamics and behavior decoding. This model includes a neural encoder with two separate modules for behavior and neural decoding, and outperforms baselines such as CEBRA and DPAD without the need for preprocessing.

**Claims And Evidence:**

This model demonstrates the benefits of integrating behavior and neural activity into one unified framework, which allows modeling the dependencies between behavior and neural dynamics. This model demonstrate superior performance in both behavior decoding and neural activity prediction. The claim is well-supported.

**Essential References Not Discussed:**

Variants of approaches that are studied in the datasets from Neural Latent Benchmark Challenges, which this work is not evaluated on, previous works such as [1][2] that decoding behaviors that not compared with.

[1] STNDT: Modeling Neural Population Activity with a Spatiotemporal Transformer.

[2] A Unified, Scalable Framework for Neural Population Decoding.

**Experimental Designs Or Analyses:**

The model is evaluated on two publicly available neural datasets, and multiple well-established baselines including CEBRA, LDA, DPAD are compared with. The model has demonstrated superior performance, while another neural latent benchmark could also be covered in this study.

**Methods And Evaluation Criteria:**

It demonstrates on two publicly available neural imaging and behavior datasets. This model demonstrated superior performance in both decoding behavior as well as predicting neural dynamics. While there are multiple related works that are not directly compared with, such as BeNeDiff, TNDM and mm-GP-VAE mentioned in the related works. Moreover, there are also variants of approaches that are studied in the datasets from Neural Latent Benchmark Challenges, which this work is not evaluated on, previous works such as [1][2] that decoding behaviors that not compared with.

[1] STNDT: Modeling Neural Population Activity with a Spatiotemporal Transformer.

[2] A Unified, Scalable Framework for Neural Population Decoding.

**Other Comments Or Suggestions:**

N/A

**Other Strengths And Weaknesses:**

The paper presents an effective solution to predict neural activities as well as behavioral decoding, it demonstrates better accuracy compared to existing approaches. While this model is not evaluated on a comprehensive Neural Latent benchmark and other existing SOTA methods.

**Questions For Authors:**

How will the model perform using neural imaging data directly, compared to using the extracted neural activities and spatial locations? Will that provide additional computational efficiency while without sacrifice of accuracy?

**Relation To Broader Scientific Literature:**

This paper worked on an important and challenging problem in brain behavior decoding. It outperforms existing well-established works such as CEBRA, and achieving higher predictive accuracy, which provides an effective method to the neuroscience community.

**Theoretical Claims:**

There are no theoretical claims or proofs.

---

> ### Author Rebuttal · Authors · 2025-04-01
>
> ### [1]: New Baselines
>
> We thank the reviewer for their constructive feedback. We agree that comparing SBIND with more baselines strengthens our contribution. In response, we have added new baseline comparisons.
>
> **Comparison with STNDT:**
>
> STNDT uses a Transformer architecture for spatiotemporal modeling of neural population spiking activity. Unlike SBIND, which processes raw images, STNDT was originally designed for Poisson-distributed spiking data. **We adapted STNDT to accept preprocessed imaging data extracted using LocaNMF**, which is a widely used approach (Wang et al, 2024). We put a Gaussian prior on LocaNMF (instead of the original Poisson prior for spikes), and trained with STNDT's original objectives (masked reconstruction and contrastive loss). Following the original work, behavior decoding was performed via ridge regression on the learned latents. We swept the number of latents for STNDT. The results (**Table B.1**), show that SBIND significantly outperforms STNDT on WFCI1 dataset.
>
>
> **Table B.1: Comparison with STNDT.**
> |Model|Num latents|Neur. R2|Beh. R2|
> |---|---|---|---|
> |STNDT|55|0.8326±0.0015|0.3529±0.0164|
> |STNDT|123|0.8408±0.0012|0.3711±0.0153|
> |STNDT|270|0.8376±0.0088|0.3951±0.0156|
> |SBIND|$n_x=16$|0.8724±0.0069|0.5059±0.0166|
>
> **Comparison with TNDM:**
>
> TNDM (Hurwitz et al., 2021) uses a sequential variational autoencoder (SVAE) to learn two sets of latent factors in spiking data with dimensionality $n_1$ and $n_2$ for behaviorally relevant and irrelevant dynamics, respectively. TNDM was originally designed for Poisson-distributed spiking data. To provide a meaningful comparison on our WFCI dataset, we adapted TNDM to accept preprocessed imaging data extracted using LocaNMF. We assumed a Gaussian distribution for these inputs and changed TNDM's neural reconstruction loss to MSE. We also swept $n_1$ and $n_2$ values for TNDM. As shown in the **newly added Table B.2**, SBIND achieves superior performance compared to this adapted TNDM in both neural prediction and behavior decoding, demonstrating the benefit of SBIND's architecture that is specifically designed for spatiotemporal image data.
>
> **Table B.2: Comparison with TNDM on WFCI1 Dataset.**
> |Model|$n_1$|$n_2$|Neur. R2|Beh. R2|
> |---|---|---|---|---|
> |TNDM|8|8|0.5442±0.0073|0.1718±0.0173|
> |TNDM|16|16|0.5464±0.0102|0.1683±0.0203|
> |TNDM|16|64|0.5022±0.0081|0.2233±0.0109|
> |SBIND|8|8|0.8724±0.0069|0.5059±0.0166|
>
> The reviewer also mentions two other methods, but direct empirical comparison to these was not feasible as described below:
>
> * **PoYo (Azabou et al., 2023)** is a foundation model for spiking activity, which builds an unsupervised, large-scale foundation model to generalize across different subjects and tasks. This is a different goal compared to SBIND’s goal of joint neural-behavioral modeling to disentangle behaviorally relevant neural dynamics. PoYo uses a tokenization scheme specific to spikes, whereas SBIND focuses on neural imaging modalities. Furthermore, the code for PoYo is not available.
>
> * **BeNeDiff:** The code for BeNeDiff is also not publicly available. BeNeDiff employs SVAEs similar to TNDM (which we now compared to). A key distinction is that SBIND processes raw imaging data, while BeNeDiff uses **LocaNMF**-preprocessed data. SBIND captures local and global spatiotemporal dependencies in images, which may be lost after preprocessing with methods such as LocaNMF.
>
> ---
> ### [2]: Neural Latent Benchmark (NLB) Comparison Request
>
> We thank the reviewer for raising this point. Unfortunately, however, all NLB datasets consist exclusively of **electrophysiological (EP) recordings**. SBIND, in contrast, is specifically designed to model the spatiotemporal grid structure of *neural imaging data* (WFCI and fUSI). Our core contribution lies in developing a method tailored to leverage this structure from raw image sequences. Therefore, while the NLB is valuable for benchmarking models designed for EP data, it is not directly applicable for evaluating the specific contributions and application of SBIND.
>
> ---
> ### [3]: Question on Raw vs Preprocessed Data
>
> This is another great point about modeling raw versus preprocessed neural images. Methods like LocaNMF extract latent components roughly corresponding to brain regions. The adapted STNDT we compared against uses learnable positional embeddings to encode spatial relationships within LocaNMF components. However, as shown in our comparison (Table B.2), this method, even with mechanisms for spatial encoding on preprocessed data, yields inferior performance compared to SBIND.
>
> In terms of computational efficiency, we acknowledge that training models on preprocessed data is generally faster than training on raw images. However, as shown in Tables 2 and 3, baselines using various preprocessing methods consistently perform worse than SBIND. Therefore, SBIND represents a trade-off that prioritizes higher predictive accuracy and the ability to learn directly from the neural images.

---

### Official Review · Reviewer_UYoU · 2025-03-13

**Overall Recommendation:** 3

**Summary:**

This work proposes SBIND, a dynamical model for neural imaging data designed to extract behaviorally relevant spatiotemporal patterns. The model mainly uses a double-RNN technique to disentangle behaviorally relevant neural dynamics from other covariates of high-dimensional neural activity. The first RNN captures the behaviorally relevant latent states, while the second RNN accounts for remaining neural dynamics. The study demonstrates that SBIND outperforms existing models on several benchmarks.

**Claims And Evidence:**

The authors shows that the proposed SBIND improves neural-behavioral decoding performance owns to effectively disentangling behavior-related and remaining neural dynamics. The results are generalizable across datasets as well as detailed ablation studies.

**Essential References Not Discussed:**

For the neural data integration:
* Extraction and Recovery of Spatio-Temporal Structure in Latent Dynamics Alignment with Diffusion Models. NeurIPS 2023, Wang et al.
* Multi-Region Markovian Gaussian Process: An Efficient Method to Discover Directional Communications Across Multiple Brain Regions. ICML 2024, Li et al.
For the neural representation learning:
* STNDT: Modeling Neural Population Activity with a Spatiotemporal Transformer. NeurIPS 2022, Le et al.
* NetFormer: An interpretable model for recovering identity and structure in neural population dynamics. 2024, Zhang et al.

**Experimental Designs Or Analyses:**

The metric of decoding for behavior prediction provides reasonable experimental validation.

**Methods And Evaluation Criteria:**

For the method, I am a bit skeptical about the neuroscience meaning of learning with neural imaging data, which is highly noisy. By comparison, may be some electrophysiology data, like Poisson spiking counts, are much more scientific meaningful. Meanwhile, the model structure of decoding neural and behavior at the same time is a bit too common actually.

**Other Comments Or Suggestions:**

Please refer to my Methods And Evaluation Criteria section.

**Other Strengths And Weaknesses:**

Please refer to my Methods And Evaluation Criteria section.

**Questions For Authors:**

No more Questions. Please refer to my Methods And Evaluation Criteria section.

**Relation To Broader Scientific Literature:**

I don't really know the contribution of this paper to broader neuroscience field given that it's only processing a certain kind of imaging data format which is hard to truly interpret.

**Theoretical Claims:**

No theoretical claims within this work.

---

> ### Author Rebuttal · Authors · 2025-04-01
>
> ### [1]: Neuroscientific meaning
> We thank the reviewer for raising this key point regarding the scientific utility of neural imaging data compared to electrophysiology (EP) data, which we will now clarify in the manuscript. Brain function relies on diverse spatial and temporal scales from single neurons to large-scale networks of neuronal populations. To understand how the brain generates behavior, we need to model neural activity across all these scales. While EP recordings measure the small-scale spiking activity of a group of single neurons, they do not measure large-scale networks that are thought to be a key basis for cognition and complex behavior (Cardin et al, 2020). Neural imaging can measure these large-scale networks by providing **large-scale spatial coverage** across cortical areas or even whole-brain access (Macé et al., 2018). As such, neural imaging data are complementary to EP data and play an increasingly central role in modern neuroscience by allowing the study of **large-scale network dynamics, functional connectivity, and mesoscale neural analyses**, which are all crucial for a full understanding of how the brain generates behavior and task performance (Musall et al., 2019; Nietz et al., 2023). Additionally, functional ultrasound imaging (fUSI) may offer a less invasive approach compared to EP for developing brain-computer interfaces (BCIs) as recently demonstrated (Griggs et al. 2024; Rabut et al., 2024).
>
> Finally, we agree with the reviewer that neural imaging and EP data have distinct statistical characteristics. Precisely for this reason, we developed SBIND to directly learn such specific spatiotemporal structure from raw neural images, to enable extracting more precise dynamical neural information. We recognize that our initial motivation for focusing on these imaging modalities could have been more clearly articulated and appreciate the reviewer’s insight.
> ### [2]: Neural-behavioral models
> First, as explained above, neural imaging data play a critical role in understanding large-scale network dynamics that are key to complex behavior and cognition. Furthermore, we demonstrated SBIND's application on two fundamentally distinct neural imaging modalities that are increasingly employed: an optical modality based on widefield calcium imaging (WFCI), and an acoustic modality based on fUSI.
>
> Second, while we agree that the general concept of joint neural-behavioral modeling has been explored, particularly for EP data, we emphasize that SBIND's novelty and contribution lie in doing so for neural imaging modalities by developing a novel architectural design tailored for the unique challenges of neural imaging data. These challenges, as highlighted in the introduction and also noted by the reviewer, include **high dimensionality, complex spatiotemporal dependencies, and prevalence of behaviorally irrelevant dynamics**. Our model differs significantly from prior joint neural-behavioral models that typically operate on lower-dimensional EP recordings that have a much smaller spatial scale than neural imaging (Sani et al., 2024), or on preprocessed time series extracted from neural imaging data (Wang et al., 2024).
>
> Finally, we demonstrated that SBIND outperforms SOTA methods, such as DPAD and CEBRA. We now also add a new baseline, STNDT (Table B.2 in our response to Reviewer 9wsc) to our results, showing SBIND’s superior performance in both behavior decoding and neural prediction compared with all baselines. Furthermore, as Reviewer z1x2 pointed out, "methods that can be applied to fUSI are relatively sparse", and to our knowledge, SBIND is the first dynamical latent state model for fUSI, whose successful application to fUSI underscores its potential to **facilitate the design of non-invasive BCIs**.
> ### [3]: References
> We thank the reviewer for bringing these relevant papers to our attention and will incorporate a discussion of them into our revised manuscript. While these represent important advancements, they differ significantly from SBIND. They all address electrophysiology data and focus on distinct goals such as cross-session latent alignment (Wang et al., 2023), modeling inter-regional communication (Li et al., 2024), capturing spatial and temporal dependencies in population activity (Le et al., 2022), or recovering interpretable inter-neuron connectivity (Zhang et al., 2024). Crucially, none are designed for joint neural-behavioral modeling, and thus they do not disentangle behaviorally relevant dynamics, nor do they employ image-specific architectural priors like SBIND's ConvRNNs with integrated self-attention. SBIND's focus thus remains distinct in leveraging the spatiotemporal structure of imaging data for robust, disentangled neural-behavioral modeling.
>
> **References:**
>
> Cardin et al., Shining a Wide Light on Large-Scale Neural Dynamics, Neuron 2020
>
> Macé et al., Whole-Brain Functional Ultrasound Imaging Reveals Brain Modules, Neuron 2018
>
> Also see Lines 452, 524, 529, 542, 559

---

> > ### Comment · Reviewer_UYoU · 2025-04-04
> >
> > Thank you for the detailed response. Please incorporate the relevant scientific meaning discussions and references from the rebuttal into the revised version of the paper. I have updated my score accordingly.

---

> > > ### Author Response · Authors · 2025-04-08
> > >
> > > We thank the reviewer for considering our response and increasing their score. We will update the manuscript by incorporating the relevant references and the discussion points regarding the scientific meaning of modeling neural images.

---

### Decision · Program_Chairs · 2025-05-01

**Decision:**

Accept (poster)

**Comment:**

The draft is about a new model of spatiotemporal relations in multivariate data for the specific domain of application of neural activity.
The scores were mostly positive but with two 3 a bit on the border, and I agree that marginal acceptance is the best recommendation here.
As it is usually the case with borderline papers, the balance of merits and weaknesses is subtle. Among the merits are the goodness of some results (but I'll never get tired of repeating the same major weakness in ICML; virtually no one supports their claims with proper inferential statistics and this draft is as flawed in this regard as many others) -this is especially critical in this paper for which the main target seems to be simply increasing accuracy at all costs-, the reasonable experimentation and the potential wide applicability. On the weaknesses, reviewer UYoU and myself coincide on the difficulty to interpret the model which in biomedical applications is often regarded as an important aspect. Also, other criticisms have been expressed such as the lack of clarity or the fact that there is a lot of previous work in the field that is oversighted here suggesting that the topic may be a bit saturated now. Indeed, the departing premise is only a half-truth; while certainly some models miss or discard spatiotemporal information but there is a wealthiness of other approaches that do consider such relations. And it does not help either that there is no clear theoretical contribution to make the draft stand out loud and clear. So in summary, taking a resultist stand the paper deserves publication but the balance could have gone the other way around easily...